# Soluble uric acid suppresses neutrophil-mediated host defense in sepsis

Qiubo Li[1,5], Juliane Anders [1,5], Kailey Flora[1], Louisa Ehreiser[1], Carolin Wendling[2], Fengjun Zhang [3,4], Liang Chang[1], Danyang Zhao[1], Li Li[1], Raimund Vogl[4], Oliver Soehnlein [3] & Stefanie Steiger [1] ✉

Neutrophils are essential for host defense and inflammation, yet their dysfunction is a hallmark of acquired immunodeficiency in kidney disease, contributing to increased susceptibility to infections such as peritonitis, sepsis, and pneumonia. We speculated that impaired renal clearance of the metabolite soluble uric acid (sUA) accounts for neutrophil dysfunction. Indeed, hyperuricemia (HU, serum UA of 9–14 mg/dL) related or unrelated to kidney disease significantly exacerbates the inflammatory immune response in mice with endotoxemia and bacterial sepsis. Despite promoting hyperinflammation, HU simultaneously impairs host defense, an effect that is partially reversible by lowering UA levels with febuxostat. We validated these findings in vitro using neutrophils or serum from healthy individuals or hyperuricemic patients with chronic kidney disease. Depleting UA partially restores neutrophil function. Mechanistically, sUA promotes neutrophil activation and degranulation but impairs phagocytosis, leading to reduced NOX2 expression independent of intracellular MPO levels. This results in diminished ROS production and defective bacterial clearance in human neutrophils. In contrast, sUA has no impact on neutrophil extracellular trap formation following exposure to LPS or *E.coli*. Together, our findings identify HU as an immunometabolic regulator that amplifies hyperinflammation, while simultaneously impairing effective host defense, suggesting that targeting UA may help to overcome acquired immunodeficiency in kidney disease.

Infections such as COVID-19, sepsis, pneumonia, hepatitis, herpes zoster, and tuberculosis affect millions of people each year and are associated with high mortality rates[1,2]. In patients with kidney disease, infectious complications represent the second leading cause of death[3,4]. Complications of severe infections can range from systemic inflammation, thrombosis, acute respiratory distress syndrome, and arterial hypotension to organ damage, including the kidney[5]. Certain infections can even trigger kidney disease, for example, anti-

neutrophil cytoplasmic antibodies-associated vasculitis, while paradoxically being prevalent in these patients[6,7]. Myeloid cells, particularly neutrophils, play a dual role in the immunopathology of infections by contributing to both hyperinflammation and immunosuppression through well-defined mechanisms including oxidative burst, bacterial killing, and neutrophil extracellular trap (NET) formation[8].

In high-risk patients where neutrophil function in host defense is impaired, clinical outcomes are worse. Diseases that interfere with the

[1]Department of Medicine IV, Division of Nephrology, Ludwig-Maximilians University Hospital, Ludwig-Maximilians University Munich, Munich, Germany. [2]Max von Pettenkofer Institute, Ludwig-Maximilians University Munich, Munich, Germany. [3]Institute of Experimental Pathology, Center for Molecular Biology of Inflammation, University of Münster, Münster, Germany. [4]Center for Information Technology (CIT), University of Münster, Münster, Germany. [5]These authors contributed equally: Qiubo Li, Juliane Anders. ✉e-mail: stefanie.steiger@med.uni-muenchen.de

immune system and contribute to neutrophil dysfunction include cancer, cardiovascular disease, diabetes, autoimmune disorders, and chronic kidney disease (CKD)[9]. Thus, patients with kidney dysfunction, especially those on dialysis, face increased infection risk and poor vaccine response due to the secondary immunodeficiency related to kidney disease (SIDKD)[1]. A number of uremic solutes/metabolites and immunoregulatory proteins, such as indoxyl sulfate, leptin, and fibroblast growth factor 23, drive uncontrolled inflammation and neutrophil dysfunction[1,10]. Neutrophil dysfunctions include decreased phagocytic capability to clear pathogens, reduced respiratory burst, altered migration[11,12], and diminished NET release and NETosis[9]. Recent evidence suggests that the metabolite soluble uric acid (sUA), namely asymptomatic hyperuricemia (HU), which most patients with CKD have, impairs β2 integrin activation and internalization/recycling, thereby altering neutrophil migration during sterile inflammation[13]. This implies an immunoregulatory role of sUA in this context. However, whether the sUA-mediated immune dysfunction in neutrophils impairs host defense to infection in kidney disease remains unclear. We hypothesized that HU affects neutrophil effector functions during endotoxemia and bacterial sepsis in CKD, whereby sUA suppresses neutrophils phagocytosis, bacterial clearance, and oxidative burst. Consistent with this hypothesis, we demonstrate that CKD-related HU and intracellular sUA drive hyperinflammation while simultaneously inhibiting neutrophil antimicrobial functions without affecting NET release, indicating that sUA-mediated neutrophil dysfunction likely contributes to SIDKD.

## Results

### Hyperuricemia amplifies the inflammatory response during LPS-induced endotoxemia and bacterial sepsis in mice

To assess whether asymptomatic HU might have immunoregulatory effects on the inflammatory response during inflammation and bacterial sepsis in vivo, we employed our transgenic mouse model of HU with and without CKD (Fig. 1a)[13–15], whereby $Glut9^{lox/lox}$ and Alb-creERT2;$Glut9^{lox/lox}$ mice were fed a chow diet enriched with inosine. Alb-creERT2;$Glut9^{lox/lox}$ mice developed asymptomatic HU in a range of 9–14 mg/dL, similar to levels found in patients with kidney disease, while $Glut9^{lox/lox}$ controls (healthy) maintained normal serum UA levels (Fig. 1b, and Supplementary Fig. S1a). Feeding Alb-creERT2;$Glut9^{lox/lox}$ mice an acidogenic diet enriched with inosine induced HU plus chronic UA nephropathy (CKD), evidenced by elevated serum UA levels (Supplementary Fig. S1a), increased plasma creatinine and BUN levels (Supplementary Fig. S1c), and reduced glomerular filtration rate (GFR, Supplementary Fig. S1b) on day 21, as well as tubular atrophy and interstitial fibrosis on day 24 (Supplementary Fig. S1d–h). In contrast, Alb-creERT2;$Glut9^{lox/lox}$ mice on a chow diet enriched with inosine developed asymptomatic HU without kidney damage (HU), as indicated by normal kidney function on day 21 (Fig. 1b, and Supplementary Figs. S1b, c), and no tubular injury and fibrosis similar to healthy control mice on day 24 (Supplementary Fig. S1d–h).

To examine the impact of asymptomatic HU (without CKD) on the response to endotoxemia, we intraperitoneally injected lipopolysaccharide (LPS) in healthy and HU mice (Fig. 1a). Twenty-four hours later, HU mice showed impaired kidney function, as indicated by elevated plasma creatinine levels (Fig. 1c), and significantly higher plasma IL-6 levels (Fig. 1d) post-LPS injection compared to controls on day 24. This aggravated inflammatory response was accompanied by increased neutrophil counts in the blood and peritoneum of HU mice compared to the healthy group (Fig. 1e–g), as well as elevated IL-6 levels in the peritoneum and higher numbers of splenic neutrophils (Fig. 1h, i). These findings indicate that asymptomatic HU amplifies the inflammatory response to LPS.

To further assess bacterial sepsis, we performed cecal ligation and puncture (CLP) in healthy and HU mice (Fig. 1j). Similar to the LPS model, HU mice displayed elevated serum UA (Fig. 1k) and plasma

creatinine (Fig. 1l) levels compared to healthy mice after CLP surgery. Interestingly, in HU mice, the bacterial growth was higher in the peritoneal lavage after CLP compared to healthy mice (Fig. 1m, n), accompanied by higher IL-6 concentrations in plasma and peritoneum (Fig. 1o, q), and increased neutrophil counts in blood, peritoneum and spleen (Fig. 1p, r). Kidney sections revealed greater vascular occlusion of intrarenal arterials in HU mice after CLP surgery (Fig. 1t).

Finally, to test whether lowering UA improves outcomes, HU mice were treated with the xanthine oxidase inhibitor febuxostat, an FDA-approved urate-lowering therapy (ULT), prior to LPS injection or CLP surgery (Supplementary Fig. S2a, h). Febuxostat treatment significantly reduced serum UA and creatinine levels in HU mice (HU + ULT) (Supplementary Fig. S2b, c, i, j), and dampened inflammation, as shown by lower IL-6 levels in plasma (Supplementary Fig. S2d, k) and peritoneum (Supplementary Fig. S2f, m), along with reduced neutrophil counts in blood (Supplementary Fig. S2e, l), peritoneum and spleen (Supplementary Fig. S2g, n) compared to untreated or sham-operated mice in both models. Together, these data indicate that asymptomatic HU without kidney dysfunction exacerbates both LPS-induced endotoxemia and bacterial sepsis by driving systemic inflammation, which can be partially mitigated by ULT.

### CKD-related HU aggravates the immune response in LPS-induced endotoxemia and gram-positive sepsis in mice

Kidney disease is associated with SIDKD and an increased infection risk[1]. To assess the putative contribution of CKD-related HU during endotoxemia and sepsis, healthy and hyperuricemic CKD (HU + CKD) mice were injected with LPS or peptidoglycan (PEP). As expected, CKD mice developed HU (Fig. 2a) and kidney dysfunction (Fig. 2b) compared to healthy mice on day 23, while LPS did not further increase these parameters in hyperuricemic CKD mice, unlike in mice with only asymptomatic HU on day 24 (Fig. 1b, c). LPS significantly increased plasma IL-6 concentrations (Fig. 2c), as well as blood neutrophil and monocyte counts (Fig. 2d, and Supplementary Fig. S3c) in healthy mice compared to PBS, an effect that was even more pronounced in hyperuricemic CKD mice. Flow cytometry analysis revealed that blood neutrophils had reduced surface expression of the age-related marker CXCR4 and CD62L (Fig. 2e) but increased expression of the chemokine receptor CXCR2 (Fig. 2f) in hyperuricemic CKD mice, suggesting CKD-related HU promotes activation and chemotaxis in neutrophils during LPS-induced endotoxemia. In the peritoneum, IL-6 and IL-1β levels were significantly higher following LPS injection (Fig. 2g, and Supplementary Fig. S3a), as well as neutrophil (Fig. 2h), macrophage (Supplementary Fig. S3b) and monocyte (Supplementary Fig. S3d) counts in hyperuricemic CKD mice as compared to healthy mice, with similar effects seen after PEP injection (Supplementary Fig. S4a–h). We did not observe UA crystals in the peritoneal fluid due to similar UA levels and pH between all groups (Supplementary Fig. S3e–g), ruling out UA crystallization as a trigger of inflammation.

Lowering UA levels with febuxostat significantly reduced serum UA and creatinine levels in hyperuricemic CKD mice (HU + CKD + ULT) (Fig. 2i–k) and dampened inflammation, as reflected by lower IL-6 levels in plasma (Fig. 2l) and peritoneum (Fig. 2p), as well as reduced neutrophil counts in blood (Fig. 2m) and peritoneum (Fig. 2q) compared to untreated mice after LPS-induced endotoxemia. Similar outcomes were observed in PEP-induced gram-positive sepsis (Supplementary Fig. S4i–q). In addition, febuxostat-treated hyperuricemic CKD mice displayed higher MFIs of CD62L and CXCR4 (Fig. 2n) but lower CXCR2 expression (Fig. 2o) on blood neutrophils with endotoxemia. Neutrophil accumulation in the spleen was increased in hyperuricemic CKD mice as compared to healthy mice during LPS-induced endotoxemia (Fig. 2r), an effect that was diminished by febuxostat treatment (Fig. 2s). Consistent results were obtained in the PEP model (Supplementary Fig. S4q). Together, our data suggest that CKD-related HU aggravates the inflammatory response to LPS-induced endotoxemia and PEP-induced

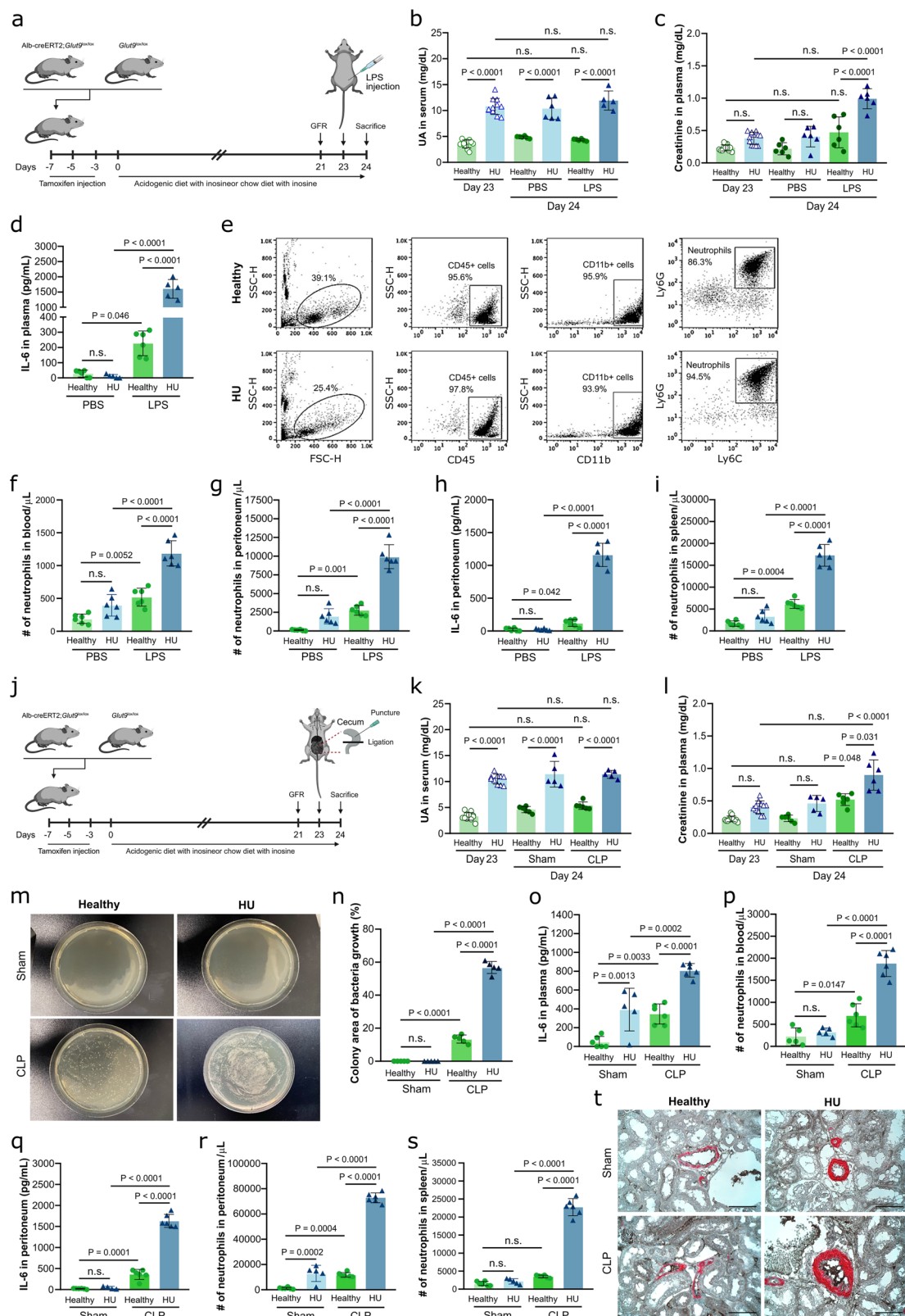

gram-positive sepsis by promoting neutrophil activation and recruitment, which can be partially mitigated by ULT.

## CKD-related hyperuricemia worsens the outcomes during bacterial sepsis in mice

To determine whether the effects observed in hyperuricemic CKD mice during LPS-induced endotoxemia and gram-positive sepsis also apply to polymicrobial sepsis, we employed the mouse model of CLP-induced sepsis. In hyperuricemic CKD (HU + CKD) mice, CLP did not further elevate serum UA (Fig. 3a) or plasma creatinine levels (Fig. 3b), while kidney function only declined in healthy mice subjected to CLP surgery compared to sham controls (Fig. 3b). Indeed, hyperuricemic CKD mice displayed significantly higher plasma IL-6 levels (Fig. 3c), increased numbers of blood neutrophils (Fig. 3d) and

**Fig. 1 | Asymptomatic hyperuricemia (HU) aggravates the immune response to LPS- and CLP-induced sepsis in mice. a** Experimental setup. Alb-creERT2;*Glut9*lox/lox mice and *Glut9*lox/lox control mice were injected intraperitoneally with tamoxifen. Both groups were fed a chow diet with inosine for 24 days. On day 21 and 24, kidney function was assessed. On day 23, mice were randomized and received either an injection of LPS or PBS into the peritoneal cavity, and were sacrificed 24 h later. Created in BioRender. Steiger, S. (2026). https://BioRender.com/ve4403h. Serum uric acid (UA) (**b**) and plasma creatinine (**c**) levels of *Glut9*lox/lox mice (healthy) and Alb-creERT2;*Glut9*lox/lox mice (hyperuricemia, HU) with inosine-rich diet after LPS or PBS injection on day 23 (*n* = 10) and 24 (*n* = 5). **b, c** One technical replicate of 5–10 biological replicates for each group. Concentrations of IL-6 measured in plasma (**d**) and peritoneum (**h**) from healthy and HU mice after LPS or PBS injection via ELISA on day 24 (*n* = 4–6, one technical replicate of 4–6 biological replicates for each group). (**e** to **g** and **i**) Gating strategy (**e**) and number (#) of neutrophils in blood (**f**), peritoneum (**g**) and spleen (**i**) per μL determined by flow cytometry (*n* = 4–6, one technical replicate of 4–6 biological replicates for each group). **j** Mice were grouped into healthy and HU group. On day 23, mice randomized and

underwent either cecal ligation and puncture (CLP) or sham surgery, and were sacrificed 24 h later. Created in BioRender. Steiger, S. (2026). https://BioRender.com/zv3eatr. Serum UA (**k**) and plasma creatinine (**l**) levels from healthy and HU mice after CLP or sham surgery on day 24 (*n* = 4–6, one technical replicate of 4–6 biological replicates for each group). Images of bacterial culture (**m**) and percentage (%) of colony area (**n**) of peritoneal wash from healthy and HU mice after CLP or Sham surgery for 12 h on LB agar plates (*n* = 4–6, one technical replicate of 4–6 biological replicates for each group). Concentrations of IL-6 measured in plasma (**o**) and peritoneum (**q**) via ELISA (*n* = 4–6, one technical replicate of 4–6 biological replicates for each group). Number (#) of neutrophils in blood (**p**), peritoneum (**r**) and spleen (**s**) per μL determined by flow cytometry (*n* = 4–6, one technical replicate of 4–6 biological replicates for each group). **t** Immunostaining performed on kidney sections for αSMA/fibrin displaying partial or complete arterial occlusions. 40x magnification, scale bar 20 μm. Data are mean ± SD. *P* values are determined by one-way ANOVA. n.s., not significant. Source data for **b–d**, **f–l**, **k**, **l**, and **n–s** are provided as a Source Data file.

monocytes (Supplementary Fig. S5c) after CLP, suggesting pronounced inflammation compared to healthy and sham-operated mice. Blood neutrophils from hyperuricemic CKD mice exhibited reduced CD62L and CXCR4 expression (Fig. 3e) but increased CXCR2 expression (Fig. 3f) compared to healthy mice after CLP, indicating increased neutrophil activation and chemotaxis. The hyperinflammatory response was associated with higher bacterial burden in the peritoneum compared to healthy mice after CLP, reflecting impaired bacterial clearance (Fig. 3g, h), accompanied with higher peritoneal IL-6 and IL-1β concentrations (Fig. 3i, and Supplementary Fig. S5a), as well as enhanced immune cell infiltration into the peritoneum (Fig. 3j, and Supplementary Fig. S5b, d) and spleen (Fig. 3k). Furthermore, CLP induced higher peritoneal MPO levels (Fig. 3l) and promoted vascular occlusion of intrarenal arteries (Fig. 3m) in hyperuricemic CKD mice compared to healthy or sham-operated mice. Notably, no UA crystals were detected in the peritoneum of hyperuricemic CKD mice (Supplementary Fig. S5e–g), confirming that the inflammation was infection-driven. Collectively, these findings demonstrate that CKD-related HU induces hyperinflammation by promoting cellular activation, cytokine release, kidney immunothrombosis and neutrophil recruitment, while simultaneously impairing pathogen clearance in bacterial sepsis.

## Urate lowering therapy diminishes the outcomes in bacterial sepsis in mice

To confirm the contribution of CKD-related HU in polymicrobial sepsis, we treated hyperuricemic CKD (HU + CKD) mice with febuxostat or vehicle prior to CLP or sham surgery (Fig. 4a). As expected, febuxostat treatment reduced serum UA and plasma creatinine levels (Fig. 4b, c) and lowered plasma IL-6 concentrations, although the latter did not reach statistical significance (Fig. 4d), in hyperuricemic CKD mice with and without CLP compared to vehicle controls. Flow cytometry analysis revealed that febuxostat increased blood neutrophil counts (Fig. 4e) without affecting their CD62L and CXCR4 expression (Fig. 4f) in hyperuricemic CKD mice, while reducing CXCR2 expression (Fig. 4g). In septic CKD mice, febuxostat also decreased bacterial load (Figs. 4h, i) and peritoneal IL-6 concentrations (Fig. 4j) while increasing neutrophil numbers in the peritoneum (Fig. 4k). Despite this increase, peritoneal neutrophils remained generally lower in hyperuricemic CKD mice (40,000 cells/μL, Fig. 4k) than in HU mice (75,000 cells/μL, Fig. 1r) with CLP, likely due to enhanced neutrophil cell death. Febuxostat further reduced peritoneal MPO levels and kidney thrombosis in septic CKD mice (Fig. 4l, m). Together, these findings suggest that febuxostat improves the outcomes during sepsis in hyperuricemic CKD mice by reducing the inflammatory response and improving host defense, as indicated by reduced bacterial burden.

## Soluble uric acid alters gene expression, activation and maturation in human neutrophils

We have previously shown that human neutrophils require the urate transporter SLC2A9 for the intracellular uptake of UA, a mechanism responsible for impaired β2 integrin activity and internalization/recycling by altering intracellular pH and cytoskeletal dynamics[13]. To elucidate the underlying molecular mechanism of how sUA affects neutrophil functions, we first examined the effects of sUA on gene expression, activation and maturation in vitro. For this, blood neutrophils were isolated from healthy individuals and stimulated for 30 min with or without 10 mg/dL sUA followed by 30 min of LPS or fMLP stimulation (Fig. 5a). Flow cytometric analysis showed that sUA increased surface expression (MFI) of CXCR2 (Fig. 5b) and CD101 (Fig. 5d), reduced CD62L (Fig. 5c), while CXCR4 remained unaffected (Fig. 5c) on neutrophils compared to medium. In addition, sUA altered cellular size and promoted degranulation in human neutrophils, as indicated by a reduced MFI of the forward scatter (FSC, Fig. 5e), the side scatter (SSC) and intracellular CD66b expression (Fig. 5g). The data suggest that sUA promotes neutrophil activation characterized by degranulation and CD62L downregulation, consistent with a more mature neutrophil phenotype, while reducing cellular size.

To look at gene expression, we isolated total RNA and performed bulk RNA-sequencing (RNA-seq) analysis of human blood neutrophils preincubated with sUA in response to LPS or fMLP. Principal-component analysis (PCA) highlighted clear segregation between activated and non-activated neutrophils in the presence or absence of sUA (Supplementary Fig. S6a). Analysis of differentially expressed genes (DEGs) uncovered a multitude of significantly upregulated inflammatory genes (e.g., *TNF*, *IL-1*, *CCL3*, *CCL4*, *CXCL1*, *CXCL3*, *NFkB*, *SOCS3*, *DUSP2*) in LPS- and fMLP-activated neutrophils compared to medium control (Supplementary Fig. S6c, d). In contrast, sUA downregulated inflammation- and adhesion-associated transcripts such as *PTGS2*, *EGR1*, *CXCL5*, *MAP3K7CL*, and *ITGB* in both non-activated and activated neutrophils (Supplementary Fig. S6b, e, f). Gene Ontology (GO) and gene set enrichment analyzes revealed that sUA alters various biological processes, including upregulation of rRNA processes, DNA replication, mitochondrial metabolism, ribosome biogenesis, energy production (ATP/NADH) (Supplementary Fig. S6g–k), suggesting altered cellular mechanisms and possibly a more aged neutrophil phenotype. Functional annotation of DEGs indicated that sUA increased NADH activity, rRNA and DNA binding, and chemokine and cytokine receptor binding/activity, but reduced transmembrane transporter activity in activated neutrophils (Supplementary Fig. S7a–c). Similarly, analysis of DEGs by cellular component also showed upregulated ribosome biogenesis, rRNA and nitric oxide processing, mitochondrial function, and NADH dehydrogenase

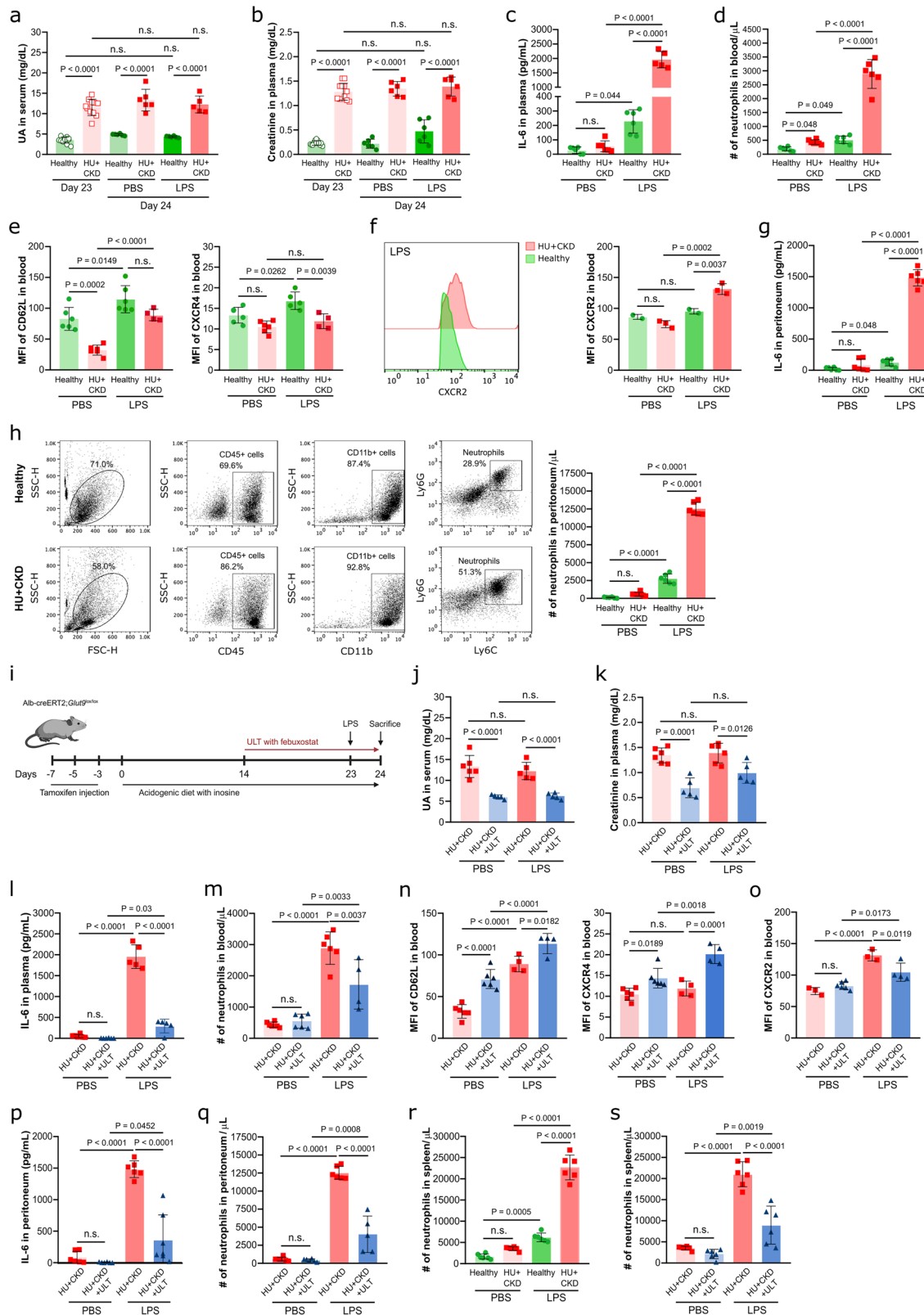

activity, but reduced integrin complex and phagocytic cup expression in sUA-treated activated neutrophils compared to controls (Supplementary Fig. S7d–f). We observed broad transcriptional reprogramming in the reactome and KEGG pathway analysis, including upregulation of rRNA modification/processes, viral mRNA translocation, receptor binding, inflammatory signaling pathways (e.g., *TLR*, *NFkB*, *p53*), ribosome biogenesis, metabolism and viral-related genes

and downregulation of platelet activation and aggregation, and NET formation (Supplementary Fig. S7g–i). These data indicate that sUA may induce a transcriptional and metabolic reprogramming in activated human neutrophils characterized by enhancing pro-inflammatory signaling, biosynthetic and metabolic activity, and altered receptor responsiveness, which may contribute to neutrophil activation but dysfunction.

**Fig. 2 | CKD-related HU aggravates the immune response to LPS-induced endotoxemia in mice.** Alb-creERT2;*Glut9*[lox/lox] mice and *Glut9*[lox/lox] control mice were injected intraperitoneally with tamoxifen. Both groups were fed either an acidogenic or chow diet enriched with inosine for 24 days. On day 21 and 24, kidney function was assessed. On day 23, mice were randomized and received either an injection of LPS or PBS into the peritoneal cavity, and were sacrificed 24 h later. Serum UA (**a**) and plasma creatinine (**b**) levels of *Glut9*[lox/lox] mice with chow diet plus inosine (healthy) and Alb-creERT2;*Glut9*[lox/lox] mice with acidogenic diet plus inosine (hyperuricemia with chronic kidney disease, HU + CKD) after LPS or PBS injection on day 23 (*n* = 10) and 24 (*n* = 5). (**a**, **b** One technical replicate of 5–10 biological replicates for each group). **c** Concentration of IL-6 measured in plasma via ELISA on day 24 (*n* = 4–6, one technical replicate of 4–6 biological replicates for each group). **d** Number (#) of neutrophils (CD45 + CD11b + Ly6G + Ly6C-) in blood per mL determined by flow cytometry (*n* = 4–6, one technical replicate of 4–6 biological replicates for each group). **e** The expression (MFI) of CD62L and CXCR4 in blood neutrophils (CD45 + CD11b + Ly6G + Ly6C-) was assessed by flow cytometry (*n* = 4–6, one technical replicate of 4–6 biological replicates for each group). **f** The expression (MFI) of CXCR2 in neutrophils (CD45 + CD11b + Ly6G + Ly6C-) was assessed in the blood by flow cytometry (*n* = 2–3, one technical replicate of 2–3 biological replicates for each group). **g** Concentrations of IL-6 measured in peritoneum via ELISA (*n* = 4–6, one technical replicate of 4–6 biological replicates for each group). **h** Neutrophil gating strategy and number (#) of neutrophils in peritoneum per μL determined by flow cytometry (*n* = 4–6, one technical replicate of 4–6 biological replicates for each group). **i** Mice were assigned to healthy and HU + CKD groups, and urate-lowering theray (ULT) with febuxostat was initiated on day 14. On day 23, mice received either an injection of LPS or PBS, and were sacrificed 24 h later. Created in BioRender. Steiger, S. (2026). https://BioRender.com/diah1lh. Serum UA (**j**) and plasma creatinine (**k**) levels from HU + CKD mice with or without febuxostat after LPS or PBS injection on day 24 (*n* = 4–6, one technical replicate of 4–6 biological replicates for each group). **l** Concentration of IL-6 measured in plasma via ELISA (*n* = 4–6, one technical replicate of 4–6 biological replicates for each group). **m** Number (#) of neutrophils in blood per μL determined by flow cytometry (*n* = 4–6, one technical replicate of 4–6 biological replicates for each group). The expression levels of CD62L and CXCR4 (**n**) and CXCR2 (**o**) in neutrophils (CD45 + CD11b + Ly6G + Ly6C-) were assessed in the blood by flow cytometry (*n* = 4–6 mice, one technical replicate of 4–6 biological replicates for each group). **p** Concentrations of IL-6 measured in peritoneum via ELISA (*n* = 4–6, one technical replicate of 4–6 biological replicates for each group). Number (#) of neutrophils (CD45 + CD11b + Ly6G + Ly6C-) in peritoneum (**q**) and spleen (**s**) per μL determined by flow cytometry (*n* = 4–6, one technical replicate of 4–6 biological replicates for each group). **r** Number (#) of neutrophils in spleen per μL determined by flow cytometry (*n* = 4–6, one technical replicate of 4–6 biological replicates for each group). Data are mean ± SD. *P* values are determined by one-way ANOVA. n.s., not significant. Source data for **a**–**h** and **j**–**s** are provided as a Source Data file.

## Soluble UA impairs phagocytosis and bacterial killing in human neutrophils

Next, we tested the effect of sUA on the ability of neutrophils to endocytose or phagocytose, produce ROS, and form NETs (Supplementary Fig. S8a). To do so, we preincubated neutrophils from healthy individuals with vehicle (control) or 10 mg/dL sUA for 30 min prior to incubation with Dextran or pathogen-associated pHrodo™ *Escherichia coli* BioParticles. Flow cytometry and fluorescence microscopy were used for quantification. Interestingly, sUA decreased the ability of neutrophils to take up Dextran compared to medium control, an effect similar to blocking phagocytosis with cytochalasin D (CytD), as indicated by a reduced MFI of Dextran (Supplementary Fig. S8b) and percentage of endocytosed Dextran (Supplementary Fig. S8c). To confirm this finding, we performed in vitro experiments with pHrodo™ *E. coli* BioParticles and found that sUA significantly inhibited the phagocytosis of pHrodo™ *E. coli* BioParticles compared to medium control, while no additional inhibitory effect was observed in combination with CytD, as indicated by fluorescence microscopy (Fig. 5h) and a lower percentage of pHrodo™ *E. coli*+ neutrophils (Fig. 5i).

To validate our in vitro findings under more physiological conditions, we preincubated healthy human neutrophils with 10 mg/dL sUA or vehicle, then assessed phagocytosis and *E. coli* killing. We found that sUA also reduced the ability of neutrophils to phagocytose *E. coli* (Fig. 5j), similar to blocking NOX with diphenyleneiodonium (DPI) and phagocytosis with cytochalasin D (CytD) but not MPO with 4-aminobenzohydrazide (4-ABAH) (Fig. 5j). The reduced phagocytic ability of neutrophils in response to sUA was associated with an impaired bacterial killing, as indicated by an increased *E. coli* viability (CFU/μl) compared to medium over 120 min (Fig. 5k). A similar trend in bacterial killing was also observed in sUA-treated LPS-activated neutrophils compared to LPS-activated neutrophils only (Fig. 5k).

In addition, we investigated the effect of intracellular sUA[13] on neutrophils by targeting the xanthine oxidase with febuxostat and depleting UA using the recombinant urate oxidase rasburicase. To do so, healthy human neutrophils were preincubated with 10 mg/dL sUA or vehicle in the presence or absence of febuxostat and rasburicase, followed by colorimetric assays, flow cytometry and phagocytosis assays. As expected, febuxostat did not reduce extracellular UA levels, which remained comparable to sUA only incubation, whereas rasburicase effectively depleted sUA from the supernatants (Fig. 5l). Exposure of neutrophils to sUA increased intracellular sUA concentrations compared with medium control (Fig. 5m). Stimulation of sUA-treated neutrophils with febuxostat did not alter intracellular UA levels, while rasburicase significantly reduced them (Fig. 5m). Flow cytometry analysis revealed that intracellular sUA did not affect the surface expression (MFI) of CXCR2 (Fig. 5n) and CXCR4 (Fig. 5p) but it decreased CD62L (Fig. 5o) and increased CD101 (Fig. 5q). Febuxostat treatment of sUA-treated unstimulated neutrophils reduced CD62L expression (Fig. 5o) and increased CXCR4 (Fig. 5p), while CD101 expression was unaffected (Fig. 5q), resembling that of sUA only treatment. Consistent with these findings, febuxostat exerted similar inhibitory effects on the phagocytic capacity of sUA-treated neutrophils as observed with sUA alone (Fig. 5r). This impairment was reversed when sUA was depleted from the medium with rasburicase, restoring phagocytic activity to levels comparable to the medium control (Fig. 5r). These data confirm that sUA alters the phenotype and phagocytic capability in neutrophils.

Next, we quantified the ability of human neutrophils to form NETs. Fluorescence microscopy revealed increased NET formation in LPS-activated neutrophils (Supplementary Fig. S9a–d), which we confirmed by measuring the release of double-stranded DNA and MPO-DNA in culture supernatants of LPS-activated neutrophils (Supplementary Fig. S9e, f). However, sUA had no effect on NET formation in LPS- and fMLP-activated neutrophils (Supplementary Fig. S9). NET formation induced by *E. coli* remained unaffected by sUA in neutrophils, as evidenced by similar expression of CitH3 and MPO (Supplementary Fig. S10a–h), while in LPS-activated neutrophils, MPO expression increased in the presence of sUA (Supplementary Fig. S10f). As expected, inhibiting phagocytosis reduced NET formation in *E. coli*-stimulated neutrophils (Supplementary Fig. S10a–h), while blocking NADPH oxidase and MPO aggravated bacterial clearance (Supplementary Fig. S10a–h). Altogether, these findings indicate that sUA impairs the ability of neutrophils to phagocytose and to kill *E. coli* while NET formation remains intact.

## Soluble UA diminishes oxidative burst in human neutrophils

Upon endocytosis/phagocytosis, neutrophils produce mitochondrial superoxide and ROS through activation of the nicotinamide adenine dinucleotide phosphate (NADPH) oxidase 2 (NOX2), which in turn mediates MPO and NE release from granules, chromatin decondensation, and is critical for pathogen killing and NET formation (Supplementary Fig. S8a)[16,17]. We speculated that intracellular sUA[13] would also affect oxidative burst in neutrophils. To test for this possibility, we first incubated neutrophils from healthy individuals with vehicle or 10 mg/dL sUA prior to stimulation with pyocyanin (PCN), a widely-used superoxide and ROS inducer. The levels of superoxide increased in

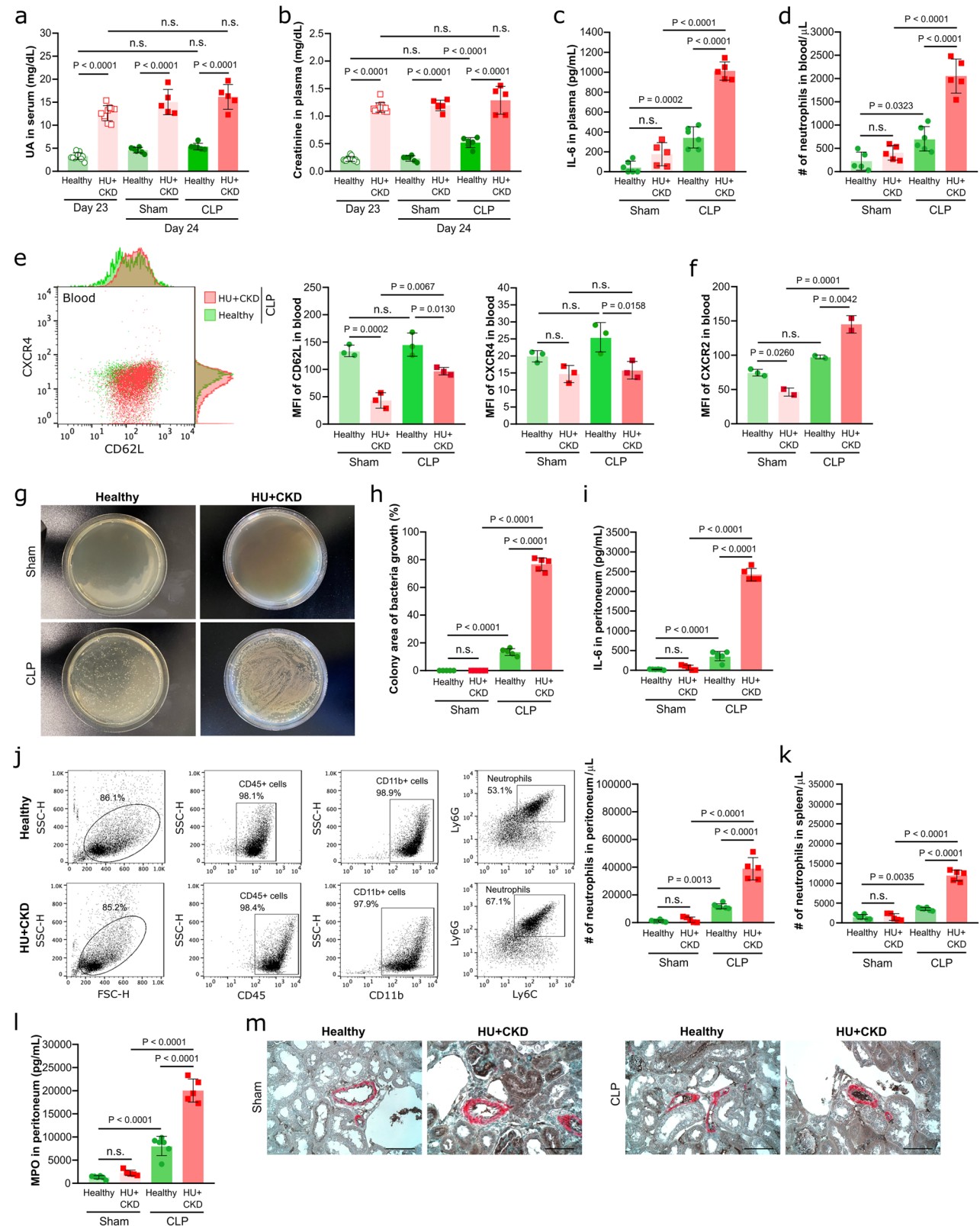

neutrophils over time, independent of treatment with sUA or PCN or in combination (Supplementary Fig. S11a, b). However, PCN induced a significant increase in ROS production compared to medium, an effect that was significantly abolished in neutrophils preincubated with sUA, as indicated by a reduced fluorescence of ROS (Supplementary Fig. S11c, d) and intracellular DHR123 in culture supernatants (Supplementary Fig. S11e, f). Stimulation of sUA-treated neutrophils with

febuxostat diminished intracellular ROS (DHR123) to similar levels as sUA-treated neutrophils only in the absence (medium) or presence of PCN (Supplementary Fig. S11g). Of note, similar to medium, the absence of sUA due to rasburicase reversed the ability of neutrophils to produce ROS (Supplementary Fig. S11g).

To further explore the impact of sUA on neutrophil function, we stimulated healthy human neutrophils with LPS or fMLP in the

**Fig. 3 | CKD-related HU aggravates the immune response to CLP-induced sepsis in mice.** Alb-creERT2;*Glut9*[lox/lox] mice and *Glut9*[lox/lox] control mice were injected intraperitoneally with tamoxifen. Both groups were fed either an acidogenic or chow diet enriched with inosine for 24 days. On day 21 and 24, kidney function was assessed. On day 23, mice were randomized and underwent either CLP or sham surgery, and were sacrificed 24 h later. Serum UA (**a**) and plasma creatinine (**b**) levels of healthy and hyperuricemic CKD (HU + CKD) mice after CLP or sham surgery on day 23 ($n = 10$) and 24 ($n = 5$). (**a**, **b** one technical replicate of 5–10 biological replicates for each group). **c** Concentrations of IL-6 measured via ELISA on day 24 ($n = 5$–6, one technical replicate of 5–6 biological replicates for each group). **d** Number (#) of neutrophils (CD45 + CD11b + Ly6G + Ly6C-) in blood per μL determined by flow cytometry ($n = 5$–6, one technical replicate of 5–6 biological replicates for each group). The expression (MFI) of CD62L and CXCR4 (**e**), and CXCR2 (**f**) in neutrophils (CD45 + CD11b + Ly6G + Ly6C-) was assessed in the blood by flow cytometry ($n = 2$–3, one technical replicate of 2–3 biological replicates for each group). Bacterial culture of peritoneal wash (**g**) and quantification of colony presence or absence of sUA. This led to an increase in superoxide area (**h**) from healthy and HU + CKD mice after CLP or sham surgery for 12 h on LB agar plates ($n = 5$–6, one technical replicate of 5–6 biological replicates for each group). **i** Concentration of IL-6 measured in peritoneum via ELISA ($n = 5$–6, one technical replicate of 5–6 biological replicates for each group). **j** Gating strategy and number (#) of neutrophils (CD45 + CD11b + Ly6G + Ly6C-) in peritoneum per μL determined by flow cytometry ($n = 5$–6, one technical replicate of 5–6 biological replicates for each group). **k** Number (#) of neutrophils (CD45 + CD11b + Ly6G + Ly6C-) in spleen per μL determined by flow cytometry ($n = 5$–6, one technical replicate of 5–6 biological replicates for each group). **l** Concentration of MPO measured in peritoneum via ELISA ($n = 5$–6, one technical replicate of 5–6 biological replicates for each group). **m** Immunostaining performed on kidneys sections for αSMA/fibrin displaying partial or complete arterial occlusions. 40× magnification, scale bar 20 μm. Data are mean ± SD. *P* values are determined by one-way ANOVA. n.s., not significant. Source data for **a**–**f** and **h**–**l** are provided as a Source Data file.

presence or absence of sUA. This led to an increase in superoxide production in LPS-treated neutrophils but not in response to fMLP (Fig. 6a). However, sUA had no effect on superoxide production (Fig. 6a). In contrast, when looking at ROS generation, stimulation of neutrophils with LPS or fMLP did not increase ROS compared to the medium control (Fig. 6b), while preincubation with sUA significantly reduced ROS production in neutrophils (Fig. 6b). This was confirmed by flow cytometric analysis showing that sUA significantly decreased intracellular ROS levels in both activated and non-activated neutrophils (Fig. 6c, d). No additional inhibitory effect of sUA on intracellular ROS generation in neutrophils was observed when blocking NADPH oxidase (NOX) with DPI and MPO with 4-ABAH (Fig. 6d). We noticed that sUA reduced mRNA expression of NOX2 (Cybb) and TLR4, whereas expression of p38a, XDN and JNK1 remained unchanged in activated neutrophils (Fig. 6e). These findings suggest that intracellular sUA may suppress ROS signaling in activated neutrophils through degranulation (Fig. 5f, g), downregulation of TLR4 and NOX2 expression (Fig. 6e), as well as reduction in xanthine oxidase activity (Fig. 6f), without affecting intracellular MPO expression (Fig. 6g).

Blocking NOX with DPI or MPO with 4-ABAH had no additional impact on intracellular MPO levels in activated and non-activated neutrophils irrespective of sUA treatment (Fig. 6g). Febuxostat treatment reduced intracellular ROS to a similar extent as sUA alone in unstimulated and LPS- or fMLP-activated neutrophils (Fig. 6h), while rasburicase reversed the suppressive effect of sUA to levels observed in control cells.

Moreover, LPS-primed neutrophils produced more superoxide in response to *E. coli* bacteria compared to unstimulated neutrophils, while sUA had no effect, similar to NOX, MPO and CytD inhibition (Fig. 6i). Interestingly, sUA significantly reduced ROS production in neutrophils in the presence or absence of *E. coli* bacteria compared to controls, an effect also seen with NOX, MPO and phagocytosis blockade (Fig. 6j).

Taken together, these data show that sUA impairs oxidative burst in human neutrophils via TLR4 and NOX2 without affecting intracellular MPO expression, suggesting an effect of intracellular sUA on ROS-dependent phagocytosis and pathogen clearance.

### Neutrophil phagocytosis and oxidative burst are impaired in hyperuricemic patients

Finally, to validate the impact of sUA on neutrophils isolated from hyperuricemic patients, we analyzed blood neutrophils from 8 patients with CKD stage G2-4 without dialysis (CKD; male/female, 6/2; mean age, 50.25 ± 21.10 years), 4 patients with CKD stage G5D on hemodialysis (ESKD; male/female, 2/2; mean age, 66.50 ± 9.77 years) and 10 healthy subjects (CKD stage 0; male/female, 6/4; mean age, 58.00 ± 3.37 years) (Fig. 7a, Tables 1 and 2). Flow cytometry revealed that neutrophils from hyperuricemic CKD and ESKD patients displayed reduced surface expression of CD62L compared to neutrophils from healthy individuals (Fig. 7c), whereas CXCR2 and CXCR4 expression levels were unchanged (Fig. 7b, d). Neutrophils from hyperuricemic patients also exhibited increased forward scatter (FSC-H, Fig. 7e) but decreased side scatter (SSC-H, Fig. 7f), indicating larger cellular size and degranulation, along with reduced intracellular expression of the granular marker CD66b in activated neutrophils from CKD patients (Fig. 7g) relative to neutrophils from healthy individuals.

Moreover, the phagocytic capacity was significantly impaired in neutrophils from CKD patients, both activated and non-activated, as shown by reduced uptake of IgG-FITC beads compared to healthy neutrophils (Fig. 7h). To test whether such effects on neutrophils are due to sUA, we isolated neutrophils from healthy subjects and preincubated the cells with serum from CKD patients or healthy individuals. The sera contained sUA levels as follows: CKD (9.76 ± 1.69 mg/dL) and healthy (5.93 ± 1.09 mg/dL). Serum from CKD patients, however, significantly decreased the ability of neutrophils from healthy individuals to phagocytose IgG-FITC beads (Fig. 7i) and to generate ROS (Fig. 7k) but not superoxide generation (Fig. 7j), an effect reversed by depleting sUA from the serum with rasburicase (Fig. 7i–k). These findings confirm that sUA impairs phagocytosis and ROS generation in neutrophils from hyperuricemic CKD patients.

## Discussion

We had hypothesized that HU affects neutrophil functions during endotoxemia and bacterial sepsis, particularly in the context of kidney disease. Our in vivo and in vitro findings support this notion, showing that CKD-related HU and intracellular sUA broadly drive hyperinflammation but at the same time suppress neutrophil effector functions, including phagocytosis, bacterial killing, and ROS generation. These effects occur without affecting NET release but appear to involve changes in cytoskeletal dynamics, cellular aging and degranulation. Collectively, our data suggest that sUA-mediated neutrophil dysfunction likely contributes to SIDKD.

Kidney disease is a known risk factor for poor outcomes and increased mortality in COVID-19[2] and other serious infections, including sepsis[18–20]. Mechanisms of SIDKD include intestinal barrier dysfunction, chronic inflammation, alterations in intestinal microbiota secretome, and immune paralysis due to impaired urinary clearance of immunoregulatory proteins and metabolites, all of which can compromise immune cell functions[1]. Neutrophil dysfunction is common in SIDKD. While HU related or unrelated to CKD is known to suppress sterile inflammation by inhibiting neutrophil recruitment as in gouty arthritis[13], our data herein reveal a distinct and context-dependent role of HU during infection. Specifically, HU exacerbates the hyperinflammatory response during endotoxemia and bacterial sepsis by

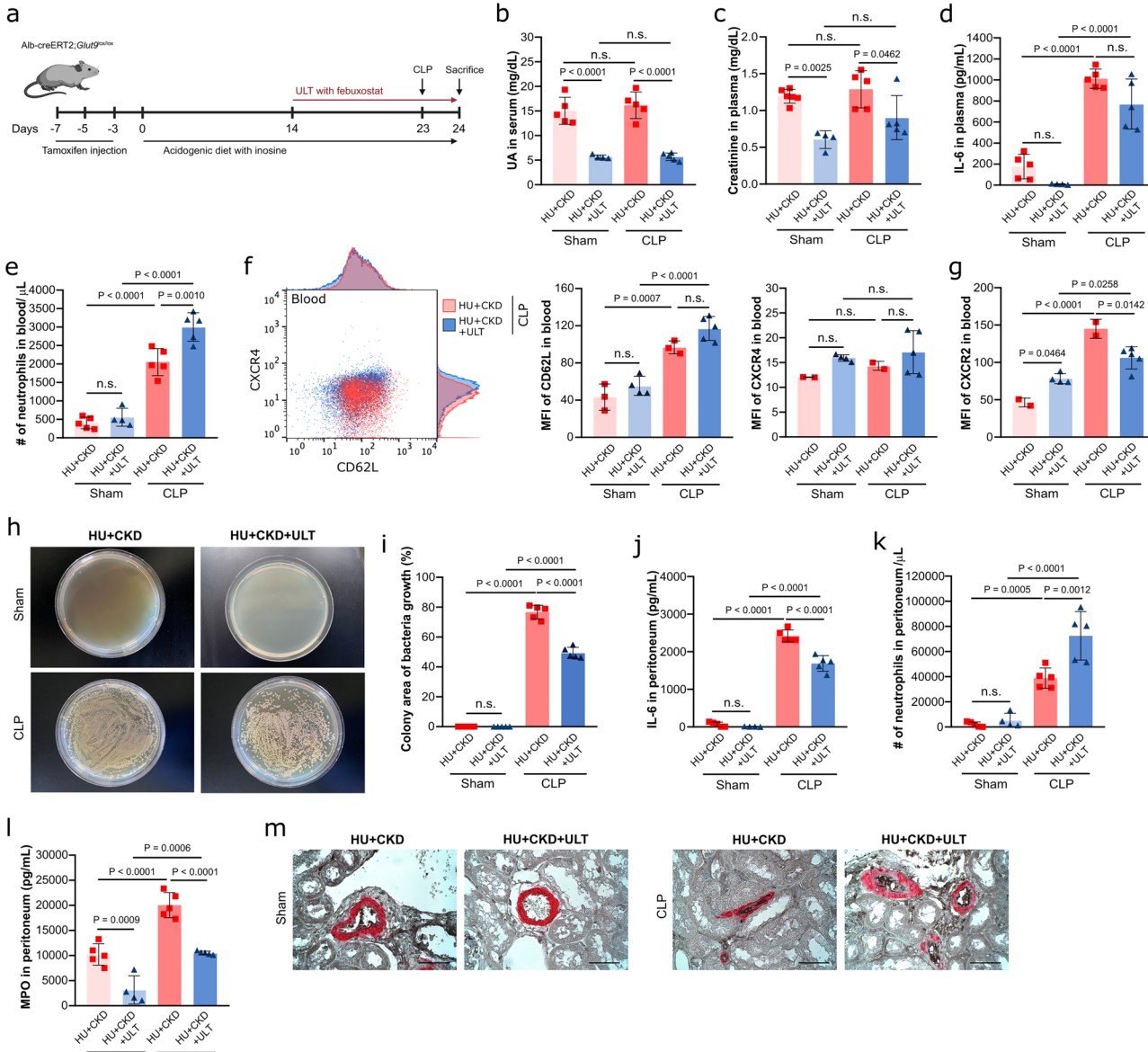

**Fig. 4 | Febuxostat treatment improves the outcomes in hyperuricemic CKD mice with CLP-induced sepsis. a** Mice were assigned to HU + CKD groups as previously described, and ULT with febuxostat was initiated on day 14 in these groups to lower serum UA levels. On day 23, mice underwent either CLP or sham surgery and were sacrificed 24 h later. Created in BioRender. Steiger, S. (2026). https://BioRender.com/4rkpxy6. Serum UA (**b**) and plasma creatinine (**c**) levels of hyperuricemic CKD (HU + CKD) mice with or without febuxostat after CLP or sham surgery on day 24 ($n = 5$). (**b**, **c** one technical replicate of 5 biological replicates for each group). **d** Concentrations of IL-6 measured in plasma via ELISA on day 24 ($n = 5$, one technical replicate of 5 biological replicates for each group). **e** Number (#) of neutrophils (CD45 + CD11b + Ly6G + Ly6C-) in blood per mL determined by flow cytometry ($n = 5$, one technical replicate of 5 biological replicates for each group). The expression (MF) of CD62L and CXCR4 (**f**), as well as of CXCR2 (**g**) in neutrophils (CD45 + CD11b + Ly6G + Ly6C-) was assessed by flow cytometry.

(**f**, **g** $n = 2$–5, one technical replicate of 2–5 biological replicates for each group). Bacterial culture of peritoneal wash (**h**) and quantification of colony area (**i**) after CLP or sham surgery for 12 h on LB agar plates. (**i**, $n = 5$, one technical replicate of 5 biological replicates for each group). **j** Concentration of IL-6 measured in peritoneum via ELISA ($n = 5$, one technical replicate of 5 biological replicates for each group). **k** Number (#) of neutrophils (CD45 + CD11b + Ly6G + Ly6C-) in peritoneum per µL determined by flow cytometry ($n = 5$, one technical replicate of 5–6 biological replicates for each group). **l** Concentration of MPO measured in peritoneum via ELISA ($n = 5$, one technical replicate of 5 biological replicates for each group). **m** Immunostaining performed on kidneys for αSMA/fibrin displaying partial or complete arterial occlusions. 40× magnification, scale bar 20 µm. Data are mean ± SD. $P$ values are determined by one-way ANOVA. n.s., not significant. Source data for **b**–**g** and **i**–**l** are provided as a Source Data file.

increasing pro-inflammatory cytokine levels, neutrophil counts and cell activation, bacterial burden, and immunothrombosis in mice. These effects were partially reversible with ULT using febuxostat. Similarly, metabolic disorders such as diabetes have been described to impair neutrophil chemotaxis, phagocytosis, and ROS production, thereby worsening bacterial sepsis outcomes[21,22]. The mechanisms linking HU to diabetes are complex and involve insulin resistance, purine intake, kidney and intestinal UA handling, reduced kidney UA

clearance, and oxidative stress. However, whether the molecular pathways through which sUA influences immune cell functions during infection overlap with or differ from those during diabetes, obesity, and autoimmune diseases remains to be determined.

To date, little is known about the impact of sUA on immune cell function during bacterial sepsis. Here, we report for the first time that intracellular sUA contributes to human neutrophil dysfunction in vitro. This aligns with previous reports showing neutrophil dysfunction in

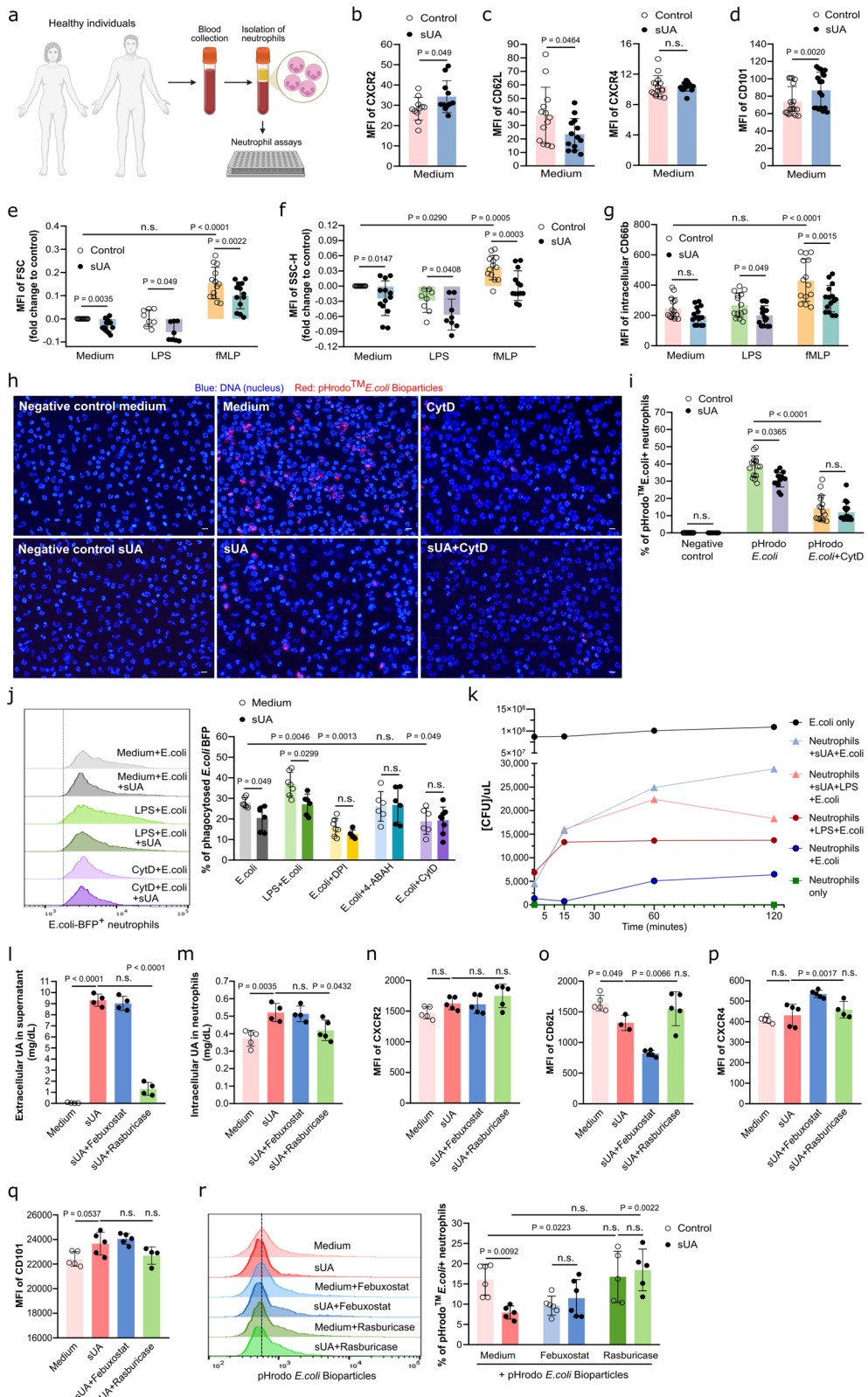

patients with kidney disease[23], pointing to a weakened immune response to clear bacterial pathogens. This impairment may result from altered cytoskeletal dynamics and reduced ROS production by down-regulating TL4 and NOX2 expression as well as decreased xanthine oxidase activity. Inflammatory mediators (e.g., CXCL8, C5a, LPS, and fMLP) are known to modulate cytoskeletal dynamics in neutrophils,

promoting their activation and migration[24,25]. Soluble UA appears to interfere with this process during sterile inflammation[13], similar to the modulatory effects of other uremia-related immunoregulatory proteins and metabolites[1,10,26,27]. However, it is possible that this effect is disease- or stimulus-dependent because we observed a higher proportion of neutrophils (CD62L[low], CXCR4[low], CXCR2[high]) in the blood of

**Fig. 5 | Soluble uric acid impairs phagocytosis and bacterial killing in human neutrophils. a** Schematic of experimental workflow including blood collection from healthy individuals, isolation of neutrophils via dextran-sedimentation, and subsequent neutrophil assays. Created in BioRender. Steiger, S. (2026). https://BioRender.com/vg41jpc. **b–d** Blood neutrophils from healthy individuals were isolated and incubated with or without sUA (10 mg/dL) for 30 min. The surface expression (MFI) of CXCR2 (**b**), CD62L and CXCR4 (**c**), and CD101 (**d**) in human neutrophils determined by flow cytometry. (**b–d**, $n = 13$, triplicates of 4 biological replicates for each group). **e–g** Healthy human neutrophils were treated with or without sUA followed by incubation with LPS (10 µg/mL) and fMLP (500 ng/mL) for 10 min. Cytoskeletal dynamics were determined via flow cytometry using the MFI of forward scatter (FSC-H) for quantifying cellular size (**e**) and the MFI of side scatter (SSC-H) (**f**) and intracellular CD66b (**g**) for quantifying granularity. (**e–g**, $n = 8$–13, duplicates or triplicates of 4 biological replicates for each group). **h** Fluorescence images of pHrodo™ *E. coli* Bioparticles™ uptake under hyperuricemic conditions compared to inhibition with cytochalasin D (CytD, 10 µM). 10x magnification, scale bar 20 µm. **i** Phagocytosis of pHrodo™ *E. coli* Bioparticles™ after neutrophils were treated with or without sUA plus CytD ($n = 15$, three technical replicates of 5 biological replicates for each group). **j** Blood neutrophils were treated with or without sUA prior to incubation with LPS for 30 min and *E. coli*, DPI, 4-ABAH, or CytD for 1 h.

Histogram and the percentage (%) of phagocytosed *E. coli* BFP was determined by flow cytometry ($n = 6$–8, two or three technical replicates of 3 biological replicates for each group). **k** Neutrophil killing of *E. coli* was determined after neutrophils were incubated with or without sUA followed by incubation with LPS and co-culture with *E. coli* in log-phase growth at a MOI of 1/20 for 2 h. At each timepoint, the co-culture was plated onto agar plates and allowed for overnight growth. Colonies were counted and the [CFU]/uL was determined using a standard curve. *E. coli* only and neutrophils only were used as controls. Extracellular UA concentrations measured in the supernatants (**l**) and intracellular sUA determined in neutrophil lysates (**m**) using a UA assay kit ($n = 4$–5). (**l, m**, one technical replicate of 4–5 biological replicates for each group). **n–q** Blood neutrophils were isolated and incubated with or without sUA in the presence or absence of febuxostat (50 µM) and rasburicase (1 µg/mL). The surface expression (MFI) of CXCR2 (**n**), CD62L (**o**), CXCR4 (**p**), and CD101 (**q**) determined by flow cytometry ($n = 3$–5). (**n–q**, one technical replicate of 3–5 biological replicates for each group). **r** Phagocytosis of pHrodo™ *E. coli* Bioparticles™ after healthy human neutrophils were treated with or without sUA in the presence or absence of febuxostat (50 µM) and rasburicase (1 µg/mL) determined by flow cytometry ($n = 5$–6, duplicates of 2–3 biological replicates for each group). Data are mean ± SD. *P* values are determined by one-or two-way ANOVA. n.s., not significant. Source data for **b–g** and **i–r** are provided as a Source Data file.

hyperuricemic mice with CKD as well as in hyperuricemic patients, suggesting a higher proportion of activated and mobilized neutrophils. While sepsis is typically associated with high numbers of young CD62L[high] and immature CXCR2[low] neutrophils in circulation[28] (known as emergency granulopoiesis), driving ROS generation and the release of pro-inflammatory cytokines and bactericidal substances like MPO[29,30], CKD-related HU and sUA appear to be rather pro-inflammatory by promoting neutrophil activation and degranulation, but at the same time impairing neutrophils' antimicrobial capacity in sepsis.

Recent studies have shown that UA can interact with MPO, a key enzyme released by activated neutrophils during degranulation and NETosis. This leads to the formation of urate-MPO reaction products and consumption of UA during oxidative burst[31–33]. UA can also act as a scavenger of peroxynitrite and other reactive nitrogen species, thereby limiting oxidative reactions mediated by MPO and NADPH oxidase[34]. These interactions may partially explain our observation of reduced ROS production without affecting superoxide and intracellular MPO expression in sUA-treated neutrophils, as urate may quench oxidants generated during the respiratory burst or modify MPO activity through redox reactions. While these effects may protect against excessive oxidative tissue injury, they could simultaneously dampen neutrophil antimicrobial capacity during sepsis, particularly in HU and CKD. Further studies are warranted to clarify the extent to which urate-MPO adducts and peroxynitrite scavenging contribute to neutrophil dysfunction in this context.

Although NETs are crucial for pathogen control, excessive NET formation may drive inflammation, immunothrombosis, and organ damage[17,35,36]. In sepsis patients, acute lung injury is common, and the presence of abundant NETs in plasma, including elevated levels of MPO-DNA complex (a marker of NETosis) and free DNA, has been linked to increased sepsis severity and mortality[37,38]. We observed an increased accumulation of mature neutrophils, elevated MPO levels, and kidney immunothrombosis in septic mice with CKD-related HU. Mature neutrophils are known to exhibit increased NET formation in mice[39]. In experimental metastatic cancer models, Peng et al. demonstrated that aged/mature neutrophils displayed increased NET formation and promoted metastasis[29]. However, proteomics analysis under normal physiological conditions revealed a loss of granular proteins in aged neutrophils, which was associated with a reduced ability to form NETs[40], suggesting that the NET-forming ability of aged neutrophils may depend on patho(physiological) states. Our in vitro data revealed that intracellular sUA promotes neutrophil degranulation isolated from healthy individuals and patients, while NADPH oxidase-dependent NET formation remained

unaffected. Further studies are needed to clarify how immature and mature neutrophils contribute to NET formation in this context and whether nonlytic, NADPH oxidase-independent, TLR4-mediated NET release, previously observed in sepsis and other infections, plays a role[17,41,42]. Evidence suggests that neutrophils from ESKD patients are more prone to apoptosis, which may impair their ability to form NETs[43,44]. Based on our data, this may be linked to neutrophil aging. These findings underscore neutrophil heterogeneity and functional diversity in both health and disease, including infections, auto-immunity, and kidney diseases[8,30,45,46].

Limitations of our study include the use of the CLP model to induce bacterial sepsis, in which the severity of sepsis can vary depending on factors such as the length of the ligated cecum, needle size, and number of punctures. These parameters, along with differences in anesthesia, surgical skills, size and number of perforations, and fecal extrusion, can vary between laboratories[47,48], making standardization, control of bacterial load and severity challenging. In our in vitro studies, we used rasburicase as a mechanistic tool (proof-of-concept) to deplete UA from the medium and serum. Of note, rasburicase converts UA into allantoin. During this degradation process, the oxidant hydrogen peroxide can also be produced as a byproduct[49]; however, we did not observe increased oxidative stress in human neutrophils in the presence of rasburicase. Moreover, in accordance with animal welfare regulations and European animal protection laws, we avoided septic shock using mortality as the primary endpoint. Alternative approaches, such as the cecal slurry or humanized bacterial infection model, may offer more standardized options[47]. Antibiotic treatment was not included in our CLP model, as the presence of fecal bacteria is essential for the induction of polymicrobial sepsis.

Overall, our findings identify an immunoregulatory role of HU in endotoxemia and bacterial sepsis, particularly in CKD, where sUA drives hyperinflammation, characterized by increased cytokine levels, neutrophil activation and recruitment, while simultaneously impairing host defense by suppressing neutrophil effector functions including phagocytosis, bacterial killing, and ROS generation. Particularly, sUA disrupts the phagocytosis-coupled NOX2-degranulation axis, where TLR4 signaling reduces phagocytic activation, leading to reduced NOX2 expression and degranulation, resulting in insufficient ROS production necessary for bacterial killing. These defects likely contribute to SIDKD. By linking CKD-related metabolic disturbances to innate immune dysfunction, our findings provide new insights into how HU may compromise immune competence during infection. Correcting HU with ULT could offer a strategy to mitigate infection-related complications in kidney disease.

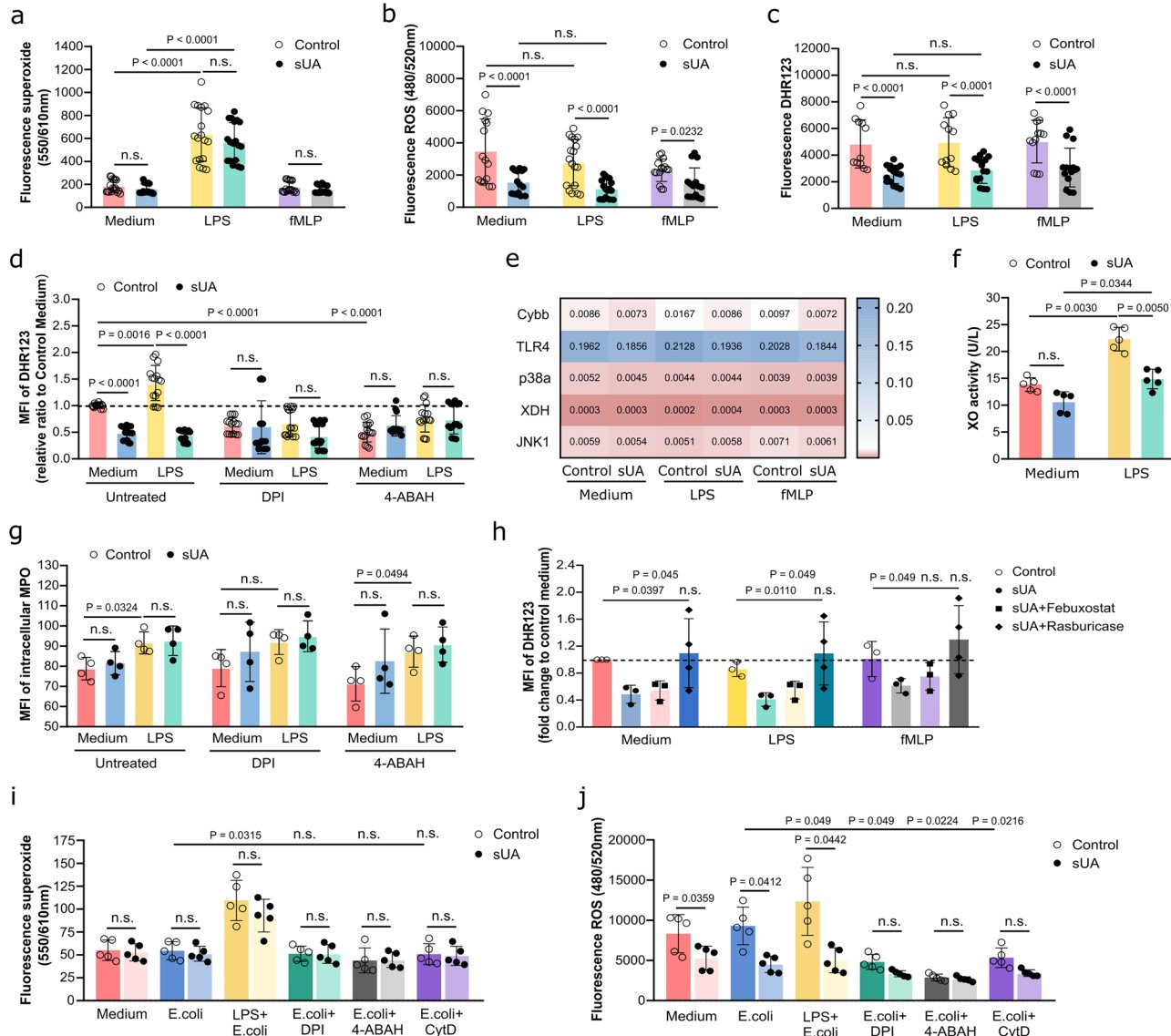

**Fig. 6 | Soluble uric acid impairs oxidative burst in human neutrophils.** Blood neutrophils from healthy individuals were isolated and incubated with or without sUA (10 mg/dL) for 30 min prior to incubation with LPS (10 μg/mL) or fMLP (500 ng/mL). **a** Superoxide production measured after healthy human blood neutrophils were treated with or without sUA (10 mg/dL) for 30 min, followed by incubation with LPS or fMLP for 40 min ($n = 18$, triplicates of 6 biological replicates for each group). **b** Total ROS was measured after neutrophils were treated with or without sUA for 30 min, followed by incubation with LPS or fMLP ($n = 18$, triplicates of 6 biological replicates for each group). **c** Fluorescence of DHR123 was measured using an ELISA reader to determine intracellular ROS after neutrophils were treated with or without sUA for 30 min, followed by incubation with LPS or fMLP for 40 min ($n = 18$, triplicates of 6 biological replicates for each group). **d** Neutrophils were treated with or without sUA as well as the inhibitors DPI and 4-ABAH followed by incubation with LPS or fMLP for 20 min ($n = 18$, triplicates of 6 biological replicates for each group). The relative ratio of the MFI of DHR123 to control medium was determined using flow cytometry and subsequent analysis in FlowJo to determine intracellular ROS. **e** Gene expression of ROS-related genes *Cybb*, *TLR4*, *p38a*, *XDH* and *JNK1* after stimulation with LPS and fMLP for 20 min ($n = 4$–7, duplicates of 2–4

biologcal replicates for each group). **f** Xanthine oxidase activity was determined in the neutrophil lysate using a xanthine oxidase assay kit ($n = 5$, one technical replicate of 5 biological replicates for each group). **g** Neutrophil inhibitors DPI and 4-ABAH were added and the MFI of intracellular MPO after stimulation with LPS for 20 min measured ($n = 4$–5, one technical replicate of 4–5 biological replicates for each group). **h** Fluorescence of DHR123 was measured using an ELISA reader to determine intracellular ROS after neutrophils were treated with or without sUA in the presence or absence of febuxostat and rasburicase, followed by incubation with LPS or fMLP ($n = 3$–4, one technical replicate of 3–4 biological replicates for each group). **i** Superoxide production measured after neutrophils were treated with or without sUA prior to incubation with LPS, DPI, 4-ABAH, or CytD, and co-culture with *E. coli* at a MOI of 1/2 ($n = 5$, one technical replicate of 5 biological replicates for each group). **j** Total ROS was measured after neutrophils were treated with or without sUA prior to incubation with LPS, DPI, 4-ABAH, or CytD, and co-culture with *E. coli* at a MOI of 1/2 ($n = 5$, one technical replicate of 5 biological replicates for each group). Data are mean ± SD. Data are mean ± SD. *P* values are determined by one-or two-way ANOVA. n.s., not significant. Source data for **a**–**j** are provided as a Source Data file.

## Methods

### Ethics statement

All animal experiments were performed in accordance with the European protection law of animal welfare and upon approval by the local government authorities Regierung von Oberbayern (ref: ROB-

55.2-2532.Vet_02-21-142) based on the European Union directive for the Protection of Animals Used for Scientific Purposes (2010/63/EU) and reported according to the Animal Research: Reporting of In Vivo Experiments (ARRIVE) guidelines[50].

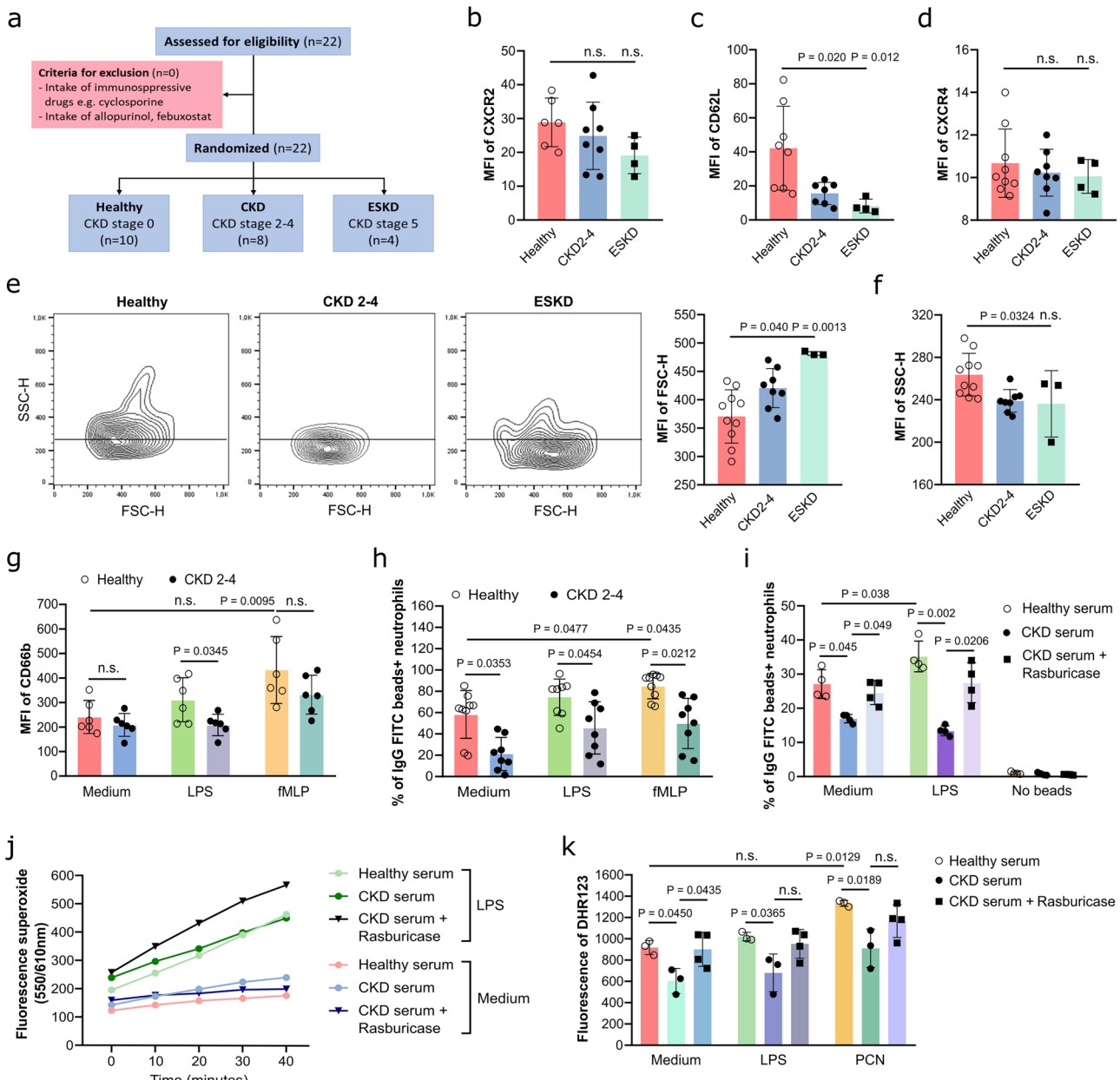

**Fig. 7 | Neutrophils from CKD patients are less able to phagocytose and produce ROS. a** Schematic of inclusion (generated with Inkscape software). Blood neutrophils from healthy individuals and patients with chronic kidney disease (CKD) and end-stage kidney disease (ESKD) were isolated and/or performed incubated with LPS (10 μg/mL) or fMLP (500 ng/mL). The expression (MFI) of CXCR2 (**b**), CD62L (**c**), and CXCR4 (**d**) on human neutrophils. (**b**–**d**, n = 4–8, one technical replicate of 4–8 biological replicates for each group). Flow cytometry was used to determine changes in cellular size (FSC-H, **e**) and granularity (SSC-H, **f**) as illustrated by dot plots and the MFI of FSC-H. (**e, f**, n = 3–10, one technical replicate of 3–10 biological replicates for each group). **g** The expression (MFI) of intracellular granular marker CD66b after stimulation with LPS or fMLP in neutrophils from healthy individuals and CKD patients (n = 6, one technical replicate of 6 biological replicates for each group). **h, i** Phagocytosis of human blood neutrophils from healthy individuals and CKD patients was assessed using IgG-FITC+ beads (n = 8–9, one

technical replicate of 8–9 biological replicates for each group) (**h**). Human blood neutrophils from healthy individuals were pre-incubated with 20% sera from healthy donors, CKD patients or CKD serum treated with rasburicase (1 μg/mL) to remove uric acid for 30 min (n = 4, one technical replicate of 4 biological replicates for each group) (**i**). After stimulation with LPS or fMLP for 10 min, phagocytosis of IgG-FITC+ beads was determined by flow cytometry. **j** Fluorescence of superoxide production in human neutrophils after stimulation with or without LPS over 40 min (n = 4, one technical replicate of 4 biological replicates for each group). **k** Intracellular ROS production (DHR123) in human neutrophils was measured as fluorescence of DHR123 after stimulation with LPS and pyocyanin (PCN) for 40 min (n = 3–4, one technical replicate of 3–4 biological replicates for each group). Data are mean ± SD. P values are determined by one-or two-way ANOVA. n.s., not significant. Source data for **b**–**k** are provided as a Source Data file.

The human study to obtain whole blood samples from healthy individuals and patients with kidney disease was approved by the local ethical review board of the medical faculty at the Ludwig-Maximilians-University (LMU) Munich (ref: 21-0532) and carried out in accordance with the Declaration of Helsinki for medical research[51]. All participants provided written informed consent.

## Animal studies

Mice were housed in groups of 5 in filter-top cages and had access to food and water. Cages, nestlets, food, and water were sterilized by autoclaving before use. Detailed information on the mouse models, including the model of HU and/or kidney dysfunction, LPS- and PEP-induced endotoxemia, and CLP-induced bacterial sepsis, is indicated below.

## Mouse models of hyperuricemia and/or kidney dysfunction

Eight-week old male and female Alb-creERT2;*Glut9*[lox/lox] mice, along with *Glut9*[lox/lox] control mice were injected intraperitoneally with tamoxifen every alternate day for 1 week to turn off *Glut9* expression in Alb-creERT2;*Glut9*[lox/lox] mice[52]. One group of Alb-creERT2;*Glut9*[lox/lox] mice was fed an acidogenic diet containing 58 kcal% fat and sucrose supplemented with 25.6 g inosine per kg (Research Diets Inc., New Brunswick, USA) to induce HU with kidney dysfunction/chronic kidney disease (HU + CKD). The second group (HU) of Alb-creERT2;*Glut9*[lox/lox] mice was given a standard chow diet with 25.6 g inosine per kg (Ssniff, Soest, Germany) to induce only HU without any renal impairment for 24 days. The *Glut9*[lox/lox] mice (healthy) received only a chow diet with 25.6 g inosine per kg (Ssniff, Soest, Germany) for 24 days, serving as the control group[14,15]. To lower serum UA levels, Alb-creERT2;*Glut9*[lox/lox] mice with HU with or without CKD were administered orally with febuxostat (3.75 mg febuxostat dissolved in 250 mL drinking water, Sigma-Aldrich) starting from day 14.

## Models of endotoxemia and bacterial sepsis

Alb-creERT2;*Glut9*[lox/lox] and *Glut9*[lox/lox] mice either on chow or acidogenic diet with inosine received intraperitoneal injections of 2 mg/kg LPS (O111:B4, Sigma-Aldrich) or 20 µg/mouse peptidoglycan (PEP, Sigma-Aldrich) or control PBS[53] on day 23 to induce LPS-induced endotoxemia or PEP-induced gram-positive sepsis. After 18 to 24 h, all groups of mice (healthy, HU, and HU + CKD) were sacrificed, and blood, peritoneal wash, kidneys, and spleens were collected for further analysis.

We chose CLP surgery to induce bacterial sepsis. For this, Alb-creERT2;*Glut9*[lox/lox] and *Glut9*[lox/lox] mice were anesthetized and a 1–2 cm midline laparotomy was performed under aseptic conditions to expose the cecum. The distal half of the cecum was ligated, followed by a single through-and-through puncture with a 26G needle. A small amount of fecal material was extruded from the perforation sites for bacterial exposure. The peritoneum was then closed, and the mice were resuscitated with a subcutaneous injection of atipamezole 2.5 mg/kg and flumazenil 0.5 mg/kg[54]. Subcutaneous injections of buprenorphine (1 mg/kg every 8 h) were given for pain control. After 24 h, all groups of mice (healthy, HU, and HU + CKD) were sacrificed.

## Human study design and procedures

The study included 8 patients with CKD-related HU without dialysis (CKD stage G2-4), 4 patients with CKD stage G5D that were on hemodialysis with a mean dialysis duration of 41 months prior to blood collection (end-stage kidney disease [ESKD]), and 10 healthy individuals without renal impairment (healthy, CKD stage G0) (Table 1). There was no significant difference in age between the groups. We excluded patients who did not meet the inclusion criteria, e.g., the use

of immunosuppressants. Blood samples from healthy individuals or patients with kidney dysfunction (CKD and ESKD) were collected in heparinized tubes (7 mL) and serum tubes (7 mL). After 30 min of coagulation at room temperature, serum was centrifuged at $3000 \times g$ for 10 min and transferred into 1.5 mL plastic Eppendorf tubes, and stored at −20 °C until analysis. Serum UA levels were measured using the Uric Acid Assay Kit (BioAssay Systems, Hayward, USA) according to the protocol provided by the manufacturer. Clinical parameters and kidney pathologies of the patients are listed in Tables 1 and 2. All participants provided written informed consent.

## Isolation of human blood neutrophils

Neutrophils were isolated as previously described[13]. Briefly, neutrophils were isolated from healthy individuals and patients with kidney dysfunction (CKD/ESKD) before dialysis using standard dextran sedimentation followed by Ficoll-Hypaque density centrifugation procedures. Neutrophils were identified by flow cytometry using the antibodies fluorescein isothiocyanate (FITC) anti-human CD11b, FITC anti-human CD15, phycoerythrin (PE) anti-human CD16 (BioLegend, Fell, Germany), and PE anti-human CD66b and allophycocyanin (APC) anti-human CD15 (both from eBioscience, Germany). Neutrophils were suspended in RPMI ($0.5 \times 10^6$ cells per 300 mL or $1 \times 10^6$ cells/mL) and counted using a hemocytometer for subsequent in vitro neutrophil assays.

## Flow cytometric analysis of murine and human samples

The peritoneal wash from mice was collected by injecting 2 mL of cold sterile phosphate-buffered saline (PBS) into the peritoneal cavity. The abdomen was gently massaged for 1–2 min, and then the same syringe was used to collect the wash, which was subsequently centrifuged at 1200 rpm for 5 min. Supernatants were collected and stored at −20 °C for further cytokine analysis. Spleens and blood were collected. Single-cell suspensions from spleens, peritoneal wash, and blood were resuspended in wash buffer and FcR blocked for 5 min before staining with the surface antibodies FITC anti-mouse Ly6G, PE anti-mouse CD11b, PE/Cy5 anti-mouse Ly6C, and APC anti-mouse CD45 (all antibodies obtained from BioLegend, Fell, Germany) for 15 min. After incubation, cells were washed with D-PBS and reconstituted in 1 mL fresh wash buffer (0.1% BSA, 0.01% sodium azide in D-PBS). Flow cytometry was performed using the BD FACSCalibur (Becton Dickinson, New Jersey, USA), and data were analyzed with the software FlowJo 10.4 (Tree Star Inc., Ashland, OR). To determine the absolute number of cells per microliter, Invitrogen AccuCheck counting beads (Thermo Fisher Scientific, PCB100, Langenselbold, Germany) were used, and absolute cell counts were calculated according to the manufacturer's instructions. Additionally, to assess neutrophil functions in blood samples, we utilized the following markers: FITC anti-mouse CD11b, PE anti-mouse CD62L, PE/Cy5 anti-mouse Ly6G, and APC anti-mouse CXCR4 as aging markers, and FITC anti-mouse Ly6G, PE anti-mouse CD11b, PE/Cy5 anti-mouse Ly6C, and APC anti-mouse CXCR2 as maturation markers (all antibodies from BioLegend). The remaining staining procedures were performed similarly as described previously.

Human blood neutrophils were isolated and incubated with or without sUA for 30 min. Afterwards, neutrophils were treated with LPS (10 µg/mL), fMLP (50 ng/mL) or PMA (25 nM) for 10 min. The reaction was stopped with 1 mL cold D-PBS, and the cells were immediately put on ice. Before staining the neutrophils for 15 min with FITC anti-human CD15, PE anti-human CD16 or APC anti-human CD11b (all antibodies obtained from BioLegend, Fell, Germany), they were washed two times with wash buffer and FcR blocked for 5 min. After incubation, cells were washed once with wash buffer and reconstituted in 200 µL fresh wash buffer (0.1% BSA, 0.01% sodium azide in D-PBS). Flow cytometry was performed using the BD FACSCalibur (Becton Dickinson, New Jersey, USA), and data were analyzed with the software FlowJo 10.8 (Tree Star Inc., Ashland, OR). To specify neutrophil functions, the

## Table 1 | Clinical and demographic characteristics of patients with kidney dysfunction

| Descriptive data | | Healthy<br>n = 10 | CKD<br>n = 8 | ESKD<br>n = 4 |
|---|---|---|---|---|
| Age (years) | Mean<br>p value | 58 ± 3.37 | 50.25 ± 21.10<br>0.409 | 66.50 ± 9.77<br>0.499 |
| Gender | Male<br>Female | 6<br>4 | 6<br>2 | 2<br>2 |
| Serum urea<br>(mg/dL) | Mean<br>p value | 29.78 ± 16.22 | 92.00 ± 50.01<br>0.0026 | 130.80 ± 32.48<br>0.0002 |
| Serum creatinine<br>(mg/dL) | Mean<br>p value | 0.72 ± 0.11 | 2.99 ± 1.95<br>0.0026 | 7.18 ± 1.07<br><0.0001 |
| Serum uric acid<br>(mg/dL) | Mean<br>p value | 5.93 ± 1.09 | 9.76 ± 1.69<br>0.0003 | 8.96 ± 2.84<br>0.0139 |
| Neutrophils per<br>mL blood (*10⁶) | Mean<br>p value | 2.06 ± 0.56 | 2.39 ± 0.87<br>0.5936 | 2.39 ± 0.92<br>0.6820 |

**Table 2 | Main human pathological diagnoses of patients with kidney dysfunction**

| Descriptive data | Healthy | CKD | ESKD |
|---|---|---|---|
| Renal pathology | None | hypertensive nephropathy, diabetic kidney disease, Alport syndrome, IgA nephropathy, drug-induced nephropathy, polycystic kidney disease | diabetic kidney disease, hypertensive nephropathy, microscopic polyangiitis |

following antibodies were used: AL488 anti-human LFA-1, PerCP anti-human CD11b, PE anti-human CD66b, APC anti-human CD101, PE anti-human CXCR2, PerCP anti-human CD62L or APC anti-human CXCR4 (all antibodies obtained from BioLegend, Fell, Germany).

### Preparation *of E. coli* for in vitro co-culture experiments
*E. coli* strains were grown from stocks stored at −80 °C in LB broth overnight at 37 °C with a shaking setting of 180 rpm, followed by a 100x dilution in LB broth for 2 h at 37 °C with shaking of 200 rpm in a Lauda Varioshake VS 150 OI to allow *E. coli* to enter log-phase growth. The concentration of *E. coli* was determined by serial dilution and absorbance measurement at 600 nm using a Tecan ELISA reader in combination with a previously established standard curve measurement. The *E. coli* strains used were the K-12 MG1655 for killing assays and ROS production, BFP-labeled K-12 MG1655 *E. coli* for phagocytosis assays, and YFP-labeled K-12 MG1655 *E. coli* for NET formation.

### Superoxide and ROS measurements
Isolated human blood neutrophils were suspended in RPMI medium without the phenol red indicator at a concentration of $1 \times 10^6$ cells/mL and incubated with or without sUA (10 mg/dL) and the ROS inhibitor N-acetyl-l-cysteine (NAC, 10 mM) for 30 min. Afterwards, neutrophils were stimulated with LPS (25 μg/mL) or neutrophil effector function inhibitors for ROS (DPI, 50 μM), phagocytosis (CytD, 10 μM), and NETosis (4-ABAH, 500 μM) for 30 min. Then nonfluorescent *E. coli* in log-phase growth was added to the culture at an MOI of 0.5 for 30 min. Additionally, neutrophils were stimulated with LPS (25 μg/mL), fMLP (500 ng/mL) or Pyocyanin (PCN, 50 μM). To assess the ROS release, the ROS-ID Total ROS/Superoxide Detection Kit (Enzo, 51010) was used. The ROS detection reagent was prepared and added to the 96-well bottom plate. ROS release was measured using an ELISA reader every 10 min for 60 to 90 min. Oxidative stress detection was measured with a fluorescein agent at excitation/emission of 490/525 nm and super-oxide detection was measured with rhodamine at excitation/emission of 550/620 nm. Between ROS measurements every 10 min, cells were placed in an incubator at 37 °C.

To measure total intracellular ROS production, activated neutrophils were incubated with the dye DHR123 (15 μM, Enzo, ENZ-52302) for 10 min. Afterwards, they were washed once with D-PBS, reconstituted in RPMI medium without phenol red and treated with ROS inducers as mentioned above. ROS release was measured using an ELISA reader at excitation/emission of 507/529 nm.

### Bacterial killing assays
Isolated human blood neutrophils were suspended in RPMI medium without the phenol red indicator at a concentration of $1 \times 10^6$ cells/mL and incubated with sUA (10 mg/dL) for 30 min. Afterwards, neutrophils were stimulated with LPS (25 μg/mL) for 30 min to observe the difference in the behavior of activated versus non-activated neutrophils killing of *E. coli*, which is relevant for comparison to different immunosuppressive or hyperactive disease states. Then non-fluorescent *E. coli* in log-phase growth were introduced to the culture at an MOI of 0.05 for 2 h. At times 0, 5, 15, 30, 60, and 120 min, a fraction of the reaction mixture was collected and plated onto agar plates (LB broth with agar, Lennox, 35 g/L). In between time points, the reaction mixture was returned to the 37 °C incubator and the plated mixture on agar plates was kept at room temperature. *E. coli* without neutrophils (*E. coli* only) was used as a control to show that *E. coli*

concentration stays consistent over time without the presence of neutrophils. After 2 h, agar plates were placed into a 37 °C incubator overnight, and the number of colonies was counted using an Inter-science Scan 50 colony counter, demonstrating the amount of colony-forming *E. coli* that survived the co-culture with neutrophils in the 2-h period. Using a standard curve, the [CFU]/uL was calculated.

### Assessment of mouse kidney injury, fibrosis and immunothrombosis
Plasma blood urea nitrogen (BUN) and creatinine (DiaSys, Holzheim, Germany) as well as serum uric acid (UA, BioAssay Systems, Hayward, USA) levels were measured using commercially available kits as per manufacturer's protocol[14]. Kidneys from mice were harvested after sacrifice. One kidney was used for flow cytometry, and the other was divided into two equal parts. One part was kept in RNA later solution at −80 °C for RNA isolation, and the second part was fixed in 4% formalin to be embedded in paraffin for histology analysis. Kidney injury and fibrosis were assessed on 2 μm thick kidney sections using periodic acid-Schiff (PAS) and Sirius red staining[14], while αSMA/fibrin staining was used to evaluate vascular occlusion of kidney arteries[36]. Images of kidney sections were taken on a Leica Microscope DMRB 301-371.011 and quantified by an independent observer.

### Glomerular filtration rate (GFR)
Mice were anesthetized with isoflurane before attaching a miniaturized imager device onto their shaved necks using a double-sided adhesive patch. This device, equipped with light-emitting diodes and a photo-diode connected to a battery, recorded the background signal of the skin for 5 min prior to intravenous injection of 150 mg/kg FITC-sinistrin[55]. Throughout the procedure, the animal remained conscious and unrestrained in a single cage with signal recording lasting approximately 1.5 h. Upon removal of the imager device, data were analyzed using MB Lab software. Glomerular filtration rate (GFR) in microliters per minute (μL/min) was calculated based on the decrease in fluorescence intensity and body weight over time, utilizing a two-compartment model along with body weight and an empirical conversion factor[56].

### Enzyme-linked immune-sorbent assay (ELISA)
Concentrations of mouse IL-6 and IL-1β in plasma and peritoneal wash extracts were measured using the mouse enzyme-linked immunoassays (ELISA) kits for IL-6 (Ray Biotech, Norcross, USA) and for IL-1β (Invitrogen, USA) according to manufacturers' protocols. The absorbance was measured on Multiskan EX reader (Thermo Electron Corporation, Germany) and Infinite 200 PRO (Tecan Austria GmbH, Austria).

### Quantification of bacterial growth from peritoneal wash of mice
To determine bacterial growth after CLP-induced sepsis in mice, 50 μL of the peritoneal wash was transferred onto the surface of an LB agar plate. Using a sterile spreader, the liquid was gently and evenly distributed across the entire agar surface. Once the liquid had spread, the agar plate was incubated at 37 °C for 12 h. After incubation, the growth of bacterial colonies was observed and recorded[57].

### Preparation of soluble uric acid for in vitro experiments
Soluble UA (sUA) was prepared as previously described[15]. Briefly, UA (Sigma-Aldrich, Taufkirchen, Germany) was solubilized in 4N NaOH.

Afterwards, the pH was adjusted to 7.4 by adding 6 N HCl and the UA stock solution (0.01 M) was filtered sterile using a 0.22 µm filter[58]. For all in vitro cell culture experiments with human blood neutrophils, we used 10 mg/dL of sUA.

## Formation and quantification of human neutrophil extracellular traps

Neutrophils from healthy individuals were pre-incubated with or without sUA (10 mg/dL) for 30 min prior to stimulation with LPS (25 µg/mL, Sigma-Aldrich, Taufkirchen, Germany), fMLP (500 ng/mL), or *E. coli* for an additional 3 h. Afterwards, neutrophils were fixed with 4% paraformaldehyde and NETs stained as previously described[59]. For immunofluorescence imaging with and without *E. coli*-YFP, neutrophils from healthy individuals were pre-incubated for 30 min with sUA (10 mg/dL), followed by a 30 min stimulation with LPS (25 nM), or DPI (50 µM), CytD (10 µM), or 4-ABAH (500 µM). Then *E. coli*-YFP at an MOI of 15 was incubated with neutrophils for 2 h to allow phagocytosis of *E. coli*. Afterwards, neutrophils were fixed with 4% paraformaldehyde, permeabilized with 0.05% Triton X-100 in D-PBS. Following permeabilization, slides were blocked using a 2% BSA buffer solution and stained using primary antibodies for MPO (mouse anti-human antibody, ab25989, Abcam) and citrullinated Histone H3 (rabbit anti-mouse and human antibody, ab5103, Abcam). Followed by several washes with D-PBS, secondary staining was completed using Alexa 488-Goat anti-rabbit IgG (111-545-144, Jackson ImmunoResearch) and Alexa 594-Goat anti-mouse (A-11032, Invitrogen). After multiple washing steps, the chambers were removed from the slides and coated in Vectashield antifade mounting medium with DAPI (Vector Laboratories, H-1200-10), and NET visualized using a Nikon Inverted Research Microscope ECLIPSE Ti2-E system.

NETs were also quantified in supernatants with the myeloperoxidase (MPO)-DNA sandwich ELISA using an anti-DNA (Roche, Germany) and anti-human MPO antibody (AbD Serotec, Raleigh, North Carolina), or with the PicoGreen dsDNA Assay Kit (Fisher Scientific, Schwerte, Germany) according to the manufacturer's instructions.

## Phagocytosis assays

Human blood neutrophils from healthy individuals ($0.25 \times 10^6$ cells per well) were pre-incubated with or without sUA (10 mg/dL) or 20% sera from healthy individuals and CKD/ESKD patients with and without rasburicase (1 µg/mL) for 30 min and then stimulated with LPS (100 ng/mL; Sigma-Aldrich, Taufkirchen, Germany) or fMLP (500 ng/mL) for an additional 15 min. To assess the phagocytic capability of neutrophils, the phagocytosis assay kit (IgG FITC) (Cayman Chemical, USA) was used. Briefly, after pre-incubation and stimulation, neutrophils were cultured at 37 °C with latex beads-rabbit IgG-FITC complex or no beads for 1 h. Afterwards, cells were washed once with assay buffer and flow cytometry analysis was performed on a FACSCalibur (BD Biosciences, Heidelberg, Germany). Data were analyzed using the software FlowJo 8.7 (Tree Star, Ashland, OR, USA).

To investigate the uptake of particles in more detail, Dextran (Invitrogen P10361) for endocytosis and pHrodo™ Red *E. coli* BioParticles™ (Invitrogen, P35361) for phagocytosis were used. Human blood neutrophils from healthy individuals ($0.15 \times 10^6$ cells per sample) were pre-incubated with or without sUA (10 mg/dL) and DPI (10 µM) for 30 min before 40 µg/mL Dextran or 50 µg/mL pHrodo™ Red *E. coli* BioParticles™ were added for 15 min. The reaction was stopped with ice-cold D-PBS, the cells were washed once with wash buffer and reconstituted in 200 µL fresh wash buffer (0.1% BSA, 0.01% sodium azide in D-PBS). Flow cytometry was immediately performed using the BD FACSCalibur (Becton Dickinson, New Jersey, USA) at Ex/Em 560/585 nm, and data were analyzed with the software FlowJo 10.8 (Tree Star Inc., Ashland, OR).

Human blood neutrophils were isolated and were pre-incubated for 30 min with soluble UA (sUA, 10 mg/dL), followed by a 30 min stimulation with LPS (25 nM) or inhibitors for ROS (DPI, 50 µM), phagocytosis (CytD, 10 µM) and NETosis (4-ABAH, 500 µM). To assess the phagocytic capability of neutrophils, *E. coli*-BFP in log-phase growth were added to the culture for 1 h at 37 °C at an MOI of 1. Afterwards, cells were washed once with cold D-PBS and centrifuged using an Eppendorf Centrifuge 5417R with a 45-30-11 rotor to remove any non-phagocytosed *E. coli*, and then resuspended in wash buffer. Different centrifugation speeds were used due to size differences: neutrophils pellet at $300 \times g$, while *E. coli* require $4000 \times g$. Flow cytometry analysis was performed to quantify phagocytosed (intracellular) *E. coli*-BFP by neutrophils on a CytoFlex flow cytometer (Beckman Coulter). Data were analyzed using the software FlowJo 8.7 (Tree Star, Ashland, OR, USA).

## RNA preparation and real-time quantitative PCR

Total RNA was isolated from mouse kidneys using the RNA extraction kit (Qiagen, Düsseldorf, Germany), following the manufacturer's instructions. The quantity of the isolated RNA was measured with a NanoDrop spectrophotometer (PEQLAB Biotechnology GMBH, Erlangen, Germany), and its integrity was assessed on 1% agarose gels. The RNA was then reverse transcribed into cDNA using Superscript II reverse transcriptase (Invitrogen, Carlsbad, CA). Real-time RT-qPCR was performed using the Light Cycler 480 system (Roche, Mannheim, Germany) with the SYBR Green I detection dye system (SYBR Green I 96 protocol LC480 Roche running program). Gene expression values were normalized to 18s rRNA as the housekeeping gene. All mouse and human primers used for amplification were obtained from Metabion (Martinsried, Germany) and are listed in Supplementary Table S1.

Human blood neutrophils from healthy individuals ($4 \times 10^6$ cells/mL) were pre-incubated with or without sUA (10 mg/dL) for 30 min before stimulation with LPS (10 µg/mL) and fMLP (500 ng/mL) for 20 min. The cells were washed once with cold D-PBS before pelleting at 2000 rpm for 10 min. RNA isolation was performed using Total RNA Purification Kit (Norgen Biotek Corp., Thorold, Canada) according to manufacture's instructions. To remove possible DNA contamination, RNA-free DNase Set (Qiagen, Düsseldorf, Germany) was used. RT-qPCR was performed as described above. For bulk RNA sequencing, samples were immediately stored at -80 °C and shipped to BGI company in China.

## Bulk RNA sequencing and analysis of human neutrophils

We extracted total RNA from human neutrophils using the Total RNA Purification Kit (Norgen Biotek Corp., Thorold, Canada). RNA concentration and integrity were determined with Agilent RNA 6000 Nano Kit using Bioanalyzer (Agilent 2100). The accession number for the RNA-seq data in GEO is GSE294890 and for SRA PRJNA1251606. RNA library preparation, quality check, sequencing and primary data processing were performed by BGI Genomics as part of their standard transcriptome sequencing service. This included mRNA enrichment using DNBSEQ Eukaryotic Strand-specific mRNA protocol, sequencing on the DNBSEQ platform with paired-end 100 bp library, and preprocessing steps, i.e., base calling, quality filtering and adapter trimming, using SOAPnuke with default parameters[60]. The resulting reads showed high quality across all samples, with Q30 scores over 94.0%. GC content ranged between 49.5 and 51.1%.

Filtered reads were aligned to the human GRCh38/hg38 reference genome (primary assemblies retrieved from Ensembl, version 113) using STAR (v2.7.10b) with paired-end mode and default parameters[61,62]. Gene quantification was conducted using featureCounts (v2.0.1)[63] against Ensembl v113 annotation of GRCh38/hg38, with recommended parameters according to official documentation. The counting matrices were merged and annotated in R for downstream analyses.

PCA of human blood neutrophils was performed using built-in R function stats::prcomp()[64]. Prior to PCA, raw count matrices were

filtered to remove genes with low expression (less than 10 counts in over 2/3 of the samples), scaled by effect size and transformed using variance stabilizing transformation (VST from DESeq2 package, with blind variance estimation), aiming for an unbiased calculation of the components.

Differential gene expression analysis of neutrophil samples was performed using DESeq2 (v1.44.0) package in R[65]. Raw data were filtered (using the same threshold as in the PCA step above), normalized/scaled, and sent to analysis using DESeq2 default model and settings. Statistical tests were performed on the following comparison pairs by passing the corresponding contrast to DESeq2::results() function: sUA vs medium, LPS vs medium, fMLP vs medium, sUA_LPS vs LPS, sUA_fMLP vs fMLP, diff(sUA_LPS, LPS) vs diff(sUA_fMLP, fMLP). Full test results were ranked and then filtered by adjusted $p$-value over 0.05 and log2(FC) over 1. MA plots were generated using ggplot2[66] and ggrepel[67] packages to visualize DEGs.

Gene set enrichment analysis (GSE) was performed and visualized via fgsea[68] and clusterProfiler[69] packages in R with default settings, plotting top 30 categories based on adjusted $p$-values against multiple functional annotation databases, including Gene Ontology (GO) for biological processes, molecular function and cellular components[70,71], KEGG[72] and Reactome pathway[73].

### Statistical analysis

Statistical analysis was performed using GraphPad Prism 8 software and data are presented as mean plus or minus standard deviation (SD). A Kolmogorov–Smirnov test, Anderson–Darling test, D'Agostino–Pearson omnibus test or Shapiro–Wilk test were used to test normality. Data were compared either by one-way analysis of variance (ANOVA) with Tukey's post-hoc test between 3 or more groups or by two-way ANOVA with Bonferroni's comparison post-hoc test when using 2 parameters with multiple groups. Differences were considered significant when $P < 0.05$. n.s. indicates not significant. All specific statistical details and the number of biological and technical replicates can be found in each corresponding figure legend.

Further information on research design is available in the Nature Portfolio Reporting Summary linked to this article.

### Reporting summary

Further information on research design is available in the Nature Portfolio Reporting Summary linked to this article.

## Data availability

The data from this study provided in the main manuscript and Supplementary Information are available in a Source data file. The Gene Expression Omnibus accession number for the RNA-seq data is GSE294890.

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

## Acknowledgements

We thank all patients and healthy volunteers involved in this study, the Dialysis Unit and Division of Nephrology for patient recruitment and assessment, and Frédéric Preitner and Bernhard Thorens from the University of Lausanne (Center for Integrative Genomics, Lausanne, Switzerland) for providing the Alb-creERT2;*Glut9*^lox/lox^ mice, Amandine Dupuis for technical support with the in vitro *E. coli* experiments, as well as Janina Mandelbaum and Anna Amifiadou for expert technical support, and Yvonne Minor and Uschi Köglsperger for animal husbandry.

Experiments to visualize neutrophils and NETs were performed using a Nikon Eclipse Ti2 microscope system (funded by DFG GZ:INST 86/1851-1 FUGG to Prof. Martha Merrow). This work was supported by grants from the Deutsche Forschungsgemeinschaft (DFG) STE2437/4-1, 4-2 and DFG TRR332 project A7 to S.S., DFG TRR332 project A2 to O.S., and DGF TRR332 INF to R.V.

## Author contributions

S.S. designed the study concept and experiments; Q.L., J.A., L.E., L.C., D.Z., L.L. and K.F. conducted experiments and analyzed data. C.W. helped with performing neutrophil killing assays and analysis. F.Z., R.V., and O.S. analyzed RNA sequencing data. S.S., Q.L. and J.A. wrote the manuscript; all contributing authors read and revised the manuscript.

## Funding

## Competing interests

The authors declare no competing interests.
