## [Transparent Peer Review file · Nature Communications]

Soluble uric acid suppresses neutrophil-mediated host defense in sepsis

Corresponding Author: Professor Stefanie Steiger

Version 0:

Reviewer comments:

Reviewer #1

(Remarks to the Author)

Li et al. investigate the ability of soluble uric acid (sUA) to alter neutrophil function in vitro and in vivo. The authors discover that sUA impairs neutrophil function, and these findings provide novel insight into our understanding of host susceptibility to infections in individuals with kidney disease. Reducing uric acid levels in vivo improved neutrophil functions. The study is comprehensive and a significant advance for the field. That said, there are many convoluted assays and approaches (e.g., multiple conditions and treatments conducted simultaneously) and many are not needed here for the main conclusions. My comments are directed largely at improving presentation and providing suggestions to streamline the manuscript.

1. I recommend removing all data related neutrophil extracellular traps (NETs) from the manuscript or placing it in the supplementary material section in its entirety. This is because the data are not convincing given the stimuli used and are a distraction from the key findings. For example, LPS is known to prolong neutrophil survival/delay apoptosis and E. coli (and phagocytosis in general) induces neutrophil apoptosis / phagocytosis-induced cells death. These process can ultimately result in secondary lysis (and non-specific NET formation) in a closed system in vitro but over an extended period of time. Thus, the findings here are seemingly at variance with previous landmark studies and an explanation is not worth the space needed. Also, the term NETosis is now used specifically for NETs that form following induction by PMA, as per the original description. Inasmuch formation of NETs can occur by non-specific osmotic lysis, no specific cell death mechanism is needed. In many reports, NETs are most likely the remains of lysed cells with bound proteins from dead neutrophils. The term NETosis should be removed unless the authors are referring to the PMA-induced phenomenon only. This entire line of experimentation and all related text could be condensed to a single line in the main text that states sUA did not elicit formation of NETs nor did it alter formation of NETs caused by any of the conditions tested. Otherwise, it is a major distraction from the primary findings.

2. The flow cytometry data showing neutrophil SSC and FSC are non-specific and should not be attributed to a specific process in the absence of testing for that process directly (e.g., degranulation, line 214). I recommend moving all of these data to supplementary material and removing any mention of these morphological changes from the main text. They are not useful.

3. Figures 1-3: please define "Before" and "After" in the legends. I assume this means before and after the indicated treatment? This is somewhat confusing on cursory read, because the "before" condition never receives/or underwent the indicated treatment.

4. Figure 4G: please clarify that the indicated cell surface molecules are measured on cells in blood and not simply in blood. Also, were these cells gated on Ly6G as in Figure 1E (for neutrophils as stated in the Figure 4G legend)? Please clarify.

5. A reduction in the gene encoding Cybb should impact the ability of mature neutrophils to produce superoxide only after extended treatments of RNA synthesis inhibitors (> 1h). But not prior to that time point b/c the cells have sufficient premade stores of the protein. This has been previously reported and tested extensively.

Minor comments/edits

Line 51. "Depleting UA partially restored neutrophil dysfunction." Should function not dysfunction.

Line 77. "In high-risk patients where neutrophil immunity to host defense is impaired." Syntax issue. Please revise to "In high-risk patients where neutrophil function in host defense is impaired, ..."

Line 91. "...whether the sUA-mediated immune dysfunction in neutrophils counteracts the host defense..." Replace counteracts with "impairs" or "alters", or something similar.

Line 222. Extend should be extent

Line 234. Remove beads from "Dextran beads". This is simply 10k MW dextran.

Line 933. "...or CKD serum eliminated from uric acid with...". Syntax issue. Please revise to "or CKD serum treated with rasburicase to remove uric acid..."

Line 926. All surface receptor expression data should refer to "on" neutrophils rather than "in" neutrophils.

Line 934. LPS of fMLP should be LPS or fMLP

Reviewer #2

(Remarks to the Author)

Patients with chronic kidney disease (CKD) often develop hyperuricemia, which may impair immune function. This study shows that elevated soluble uric acid (sUA) suppresses neutrophil phagocytosis, ROS, and bacterial killing. Using mouse models of endotoxemia and CLP-induced sepsis, they demonstrate that hyperuricemia, both with and without CKD, worsens infection outcomes by increasing systemic inflammation, peritoneal bacterial load, and kidney injury. CKD-associated hyperuricemia further elevated neutrophil counts, size, degranulation, cytokine levels, and renal immunothrombosis. Treatment with the uric acid-lowering drug febuxostat reduced sUA levels, kidney dysfunction, peritoneal bacterial burden, and inflammatory markers across multiple compartments. In human studies, neutrophils from CKD and end-stage kidney disease patients showed reduced phagocytic activity and ROS. Exposing healthy neutrophils to serum from CKD patients reproduced this dysfunction, which was partially reversed by rasburicase, an enzymatic uricase. These findings suggest that sUA contributes to neutrophil dysfunction and impaired host defense in CKD and that urate-lowering therapy may help restore immune competence.

Comments:

- Although the models are well described in the methods, a concise explanation in the results (i.e., which diets induce HU vs CKD+HU, with or without kidney injury) would improve readability. Clarify what "before" and "after" mean in this context.
- While it's appreciated that the authors distinguish bacterial sepsis from endotoxemia, note that CLP is variable and lacks reproducibility. Consider briefly mentioning alternative sepsis models (e.g., cecal slurry, humanized bacterial infection models). Also, the absence of antibiotics in the CLP model reduces clinical relevance and should be acknowledged.
- The term "endotoxemia" typically refers to LPS-induced inflammation, not peptidoglycan. Please clarify whether PEP-induced inflammation is better described as a sterile immune challenge or inflammatory mimic of gram-positive sepsis, rather than "endotoxemia."
- Inclusion of survival data after febuxostat treatment would strengthen the translational impact.
- Using side scatter (SSC) to assess degranulation is not optimal. More specific markers (e.g., CD63, CD66b) or functional granule assays would provide better mechanistic insight.
- The RNA-seq section lacks functional interpretation. Rather than stating that sUA alters gene expression and cell size, specify which pathways are affected (e.g., downregulation of migration, calcium signaling, cytoskeletal remodeling, bactericidal responses). Distinguish between phenotypic and mechanistic observations.
- Overall, the study presents strong phenotypic data (impaired ROS and phagocytosis) but lacks deeper mechanistic exploration, this should be acknowledged or clarified.
- The UA levels in patient samples are reported in Table 1 but not clearly stated in the results. Please summarize patient UA levels in the results and state how much blood was collected for these assays.
- The rationale for using febuxostat in mice and rasburicase in human neutrophil assays should be explained (pharmacologic differences, in vitro applicability).
- Are circulating neutrophil numbers different between hyperuricemic patients and healthy individuals? If not assessed, this limitation should be acknowledged.
- The manuscript states reduced CXCR2 and increased aged neutrophils in CKD/ESKD patients, but the data show no statistical difference. These claims should be corrected for accuracy.
- The killing assay methods are insufficiently detailed. Clarify:
 - o Were you measuring extracellular bacteria only, or total (extra + intracellular)?
 - o How was phagocytosis distinguished from killing?
 - o How was bacterial uptake confirmed prior to measuring killing?
 - o Why were neutrophils stimulated with LPS before E. coli exposure?
 - o Could the neutrophil response to LPS vs E. coli differ by disease state?
- In Fig 6A, the CFU values (~10⁸/μL) appear extremely high given the reported MOI of 0.05 per 10⁶ neutrophils. This discrepancy should be rechecked for accuracy or clarified in the methods.
- In Figure 1B, it is unclear what "before" and "after" refer to—presumably before and after the acidogenic diet. This should

be clarified both in the figure legend and results text.

- In Figure 1C, 1K–L, serum uric acid is elevated in both “before” and “after” LPS groups, but only the “after” group develops kidney dysfunction. This discrepancy needs to be explained—possibly related to cumulative exposure or duration?
- Figure 1S kidney section results are labeled incorrectly in the results text; the figure reference should be corrected.
- Figure 2E description of neutrophil markers is inaccurate. Results show no change in CD62L, reduced CXCR4, and increased CXCR2 in CKD mice during endotoxemia, not as stated in the text. Please revise for accuracy and interpret significance appropriately.
- In Figure 1M, bacterial clearance is shown to be impaired in CKD mice, but CFUs should be quantified to substantiate the claim.
- Supplemental Figure 1, mouse models are described clearly here, but a brief summary should be included in the main results to aid clarity.
- Evidence of kidney injury in HU mice (e.g., creatinine, fibrosis) is currently only in supplemental figures, this should be moved to the main figures given its importance.
- Figure 7/Table 1, CKD and ESKD patients in Table 1 have serum UA levels below the defined hyperuricemia range (9–14 mg/dL). Consider addressing this inconsistency and clarifying whether any of these patients had infections, to support the clinical relevance of findings.

This study addresses an important and understudied aspect of immune dysfunction in CKD-related hyperuricemia, offering compelling evidence that soluble uric acid impairs neutrophil-mediated host defense during sepsis. The data are promising and the translational relevance is high. However, the manuscript is significantly hindered by unclear figure legends, inconsistent and sometimes inaccurate interpretation of results, incomplete methodological descriptions, weak mechanistic insight, and underdeveloped or clinically limited models (e.g., lack of antibiotics, questionable hyperuricemia definitions in patient cohorts). These issues collectively detract from the clarity, rigor, and overall impact of the work. Addressing these concerns will be essential to strengthen the conclusions and improve the manuscript’s contribution to the field.

Reviewer #3

(Remarks to the Author)

The authors present evidence that hyperuricemia may exacerbate inflammation due to infection or endotoxemia by altering the ability of neutrophils to kill bacteria. Specifically, they use an interesting model of hyperuricemia due to liver-specific knockout of Glut9 in which hyperuricemia develops in response to inosine but CKD only develops if the diet is acidified (thought to be related to crystalluria or intrarenal crystal). They first show that either LPS or cecal puncture/sepsis worsens AKI and bacterial cultures in the peritoneum in hyperuricemic mice compared to healthy controls. They also show that their hyperuricemia CKD model is also exacerbated by LPS or cecal puncture, and in these models the administration of febuxostat is associated with less sepsis and inflammation, and better creatinine. They then show that soluble uric acid affects neutrophil phagocytosis and oxidant release and impairs bacterial killing, but not NET formation. Finally, they take human neutrophils from healthy subjects or with CKD and show that CKD is associated with higher uric acid levels and worse neutrophil function, and that reducing uric acid with rasburicase can improve neutrophil function.

Overall this is very nice work, and the authors are to be congratulated. However, there are some very important experiments missing.

1. The most important study is to determine if hyperuricemia worsens sterile inflammation. Indeed, the authors seem to imply uric acid might be protective here, but there is no convincing evidence presented. I suggest the authors use a model such as carriage-induced peritonitis. I believe hyperuricemia will make kidney disease worse, as determined with and without febuxostat.
2. The authors also show that asymptomatic hyperuricemia is not so asymptomatic if endotoxin or sepsis occurs, and this is associated with worsening creatinine (AKI) and neutrophilia, etc. Again, this should be shown to be due to the uric acid by doing studies with febuxostat.
3. There is some data showing histology that hyperuricemia is affecting SMA or fibrin deposition. Can the authors do specific staining of the kidneys for inflammation, fibrosis, SMA and fibrin? We need to know how the kidneys are being affected.
4. The model of hyperuricemia is quite unique and many investigators do not believe this is a good model as it affects liver processing of uric acid. It might be ideal to show at least in one other model of hyperuricemia that these findings hold.
5. Finally, incubation studies with rasburicase are quite problematic as rasburicase releases oxidants on reducing urate. These studies should be deleted. The human studies would have been much stronger by altering uric acid with XO inhibitors and documenting reduced XO activity and intracellular urate levels.
6. Uric acid is also known to form MPO-related adducts and to block peroxynitrite, and how this plays into the results shown is not so clear.

Version 1:

Reviewer comments:

Reviewer #2

(Remarks to the Author)

The authors have substantially revised and improved the manuscript according to reviewer comments.

Reviewer #3

(Remarks to the Author)

The main finding of the paper is that infection or LPS can induce a more prominent inflammatory response in hyperuricemic mice associated with renal dysfunction and that this can be improved by lowering the serum uric acid with febuxostat. While superficially the enhanced inflammatory response associated with hyperuricemia would suggest uric acid is promoting inflammation, the authors make the case that this is the reverse, based on in vitro studies that uric acid is impairing the killing of the bacteria due to effects on phagocytosis and oxidative burst.

There remain several key issues that I feel should be addressed.

1. The authors actually have mixed data with uric acid showing some proinflammatory and anti-inflammatory effects, so perhaps dysfunctional is a better term.

For example, in both in vitro and in vivo studies, uric acid is associated with enhanced degranulation, which is a principle sign of neutrophil activation. They actually show data that extracellular MPO levels are high. We need to know more about this- are proteases and defensins being released in addition to MPO? This is a classic proinflammatory effect. Chemotaxis was also not measured. This should also be done.

In contrast, the reduction in oxidants presented is most likely from quenching of peroxynitrite with uric acid. This would impair killing.

Despite hyperuricemic PMNs showing impaired killing, they may also be proinflammatory. One interesting finding by the authors is that the neutrophils are aged. One explanation could be that hyperuricemia may act as a DAMP to stimulate low grade activation of PMNs, which are relatively spent and exhausted and may not mount effective killing but are contributing to systemic inflammation. It might be interesting to do a time response to see how long it takes for a healthy PMN to change its phenotype to an aged neutrophil (CXCR4) in the presence or absence of uric acid. Then a comparison of neutrophil inflammatory responses (chemotaxis, inflammatory cytokine release) in the two states could be compared.

2. The authors hypothesize that the worse inflammation with hyperuricemia is because the uric acid is impairing bacterial clearance, so the inflammation mounts. In this scenario, the animals should die sooner from severe infection. However, the authors state that survival studies cannot be done so this important study cannot be done. However, if the authors are correct, and the proinflammatory response is from a higher bacterial load, then treating animals with antibiotics would be expected to fully correct the increased inflammatory response in hyperuricemic animals. However, if hyperuricemia is causative for a greater inflammatory response, then we should see the opposite. One way to do this is to initiate antibiotics at the time the cecal ligation model is induced, and to compare results.

3. The other way to test this is with a true sterile inflammation model. I do not think the prior studies address this. Carageenan induced peritonitis only takes a few hours to induce and would be perfect (see PMID: 26786873). I would recommend doing this two ways.

One way is to create the hyperuricemic model with inosine for 23 days and then do an acute carrageenan peritonitis. The other would be to give the inosine one day before the carrageenan. This way one can get an idea for a chronic hyperuricemic state (such as in the current models presented) and a more acute model.

4. The human PMN studies are problematic since kidney dysfunction and diabetes are known to induce PMN dysfunction, and so evidence that this is due to hyperuricemia vs other mechanisms affecting neutrophils. There are CKD patients with normal uric acid levels (only 50 percent are hyperuricemic at time of initiation of dialysis) so the right control would be nondiabetic CKD with similar GFR but different uric acid levels. The authors also do not understand how rasburicase works. When it degrades uric acid, it produces oxidants (primarily hydrogen peroxide) so it is not a good way for testing uric acid in an in vitro system. I would probably delete this whole section unless it is done with the correct controls.

Version 2:

Reviewer comments:

Reviewer #2

(Remarks to the Author)

The authors have more than adequately addressed reviewer concerns.

Thank you for giving us the opportunity to revise our manuscript (NCOMMS-25-37941A) entitled “Soluble uric acid suppresses neutrophil-mediated host defense in sepsis”, which you can find enclosed. We have addressed all comments from the reviewers’ point-by-point as outlined below and included additional data to strengthen our novel concept of soluble uric acid (sUA) acts as an endogenous regulator of innate immunity during sepsis by impairing neutrophil effector functions and potentially explains the increased infection risk in patients with kidney disease.

Reviewer Comments:

Reviewer #1 (Remarks to the Author):

Li et al. investigate the ability of soluble uric acid (sUA) to alter neutrophil function in vitro and in vivo. The authors discover that sUA impairs neutrophil function, and these findings provide novel insight into our understanding of host susceptibility to infections in individuals with kidney disease. Reducing uric acid levels in vivo improved neutrophil functions. The study is comprehensive and a significant advance for the field. That said, there are many convoluted assays and approaches (e.g., multiple conditions and treatments conducted simultaneously) and many are not needed here for the main conclusions. My comments are directed largely at improving presentation and providing suggestions to streamline the manuscript.

Reply: *We thank the reviewer for his/her positive evaluation of our manuscript.*

1. I recommend removing all data related neutrophil extracellular traps (NETs) from the manuscript or placing it in the supplementary material section in its entirety. This is because the data are not convincing given the stimuli used and are a distraction from the key findings. For example, LPS is known to prolong neutrophil survival/delay apoptosis and E. coli (and phagocytosis in general) induces neutrophil apoptosis / phagocytosis-induced cells death. These process can ultimately result in secondary lysis (and non-specific NET formation) in a closed system in vitro but over an extended period of time. Thus, the findings here are seemingly at variance with previous landmark studies and an explanation is not worth the space needed. Also, the term NETosis is now used specifically for NETs that form following induction by PMA, as per the original description. Inasmuch formation of NETs can occur by non-specific osmotic lysis, no specific cell death mechanism is needed. In many reports, NETs are most likely the remains of lysed cells with bound proteins from dead neutrophils. The term NETosis should be removed unless the authors are referring to the PMA-induced phenomenon only. This entire line of experimentation and all related text could be condensed to a single line in the main text that states sUA did not elicit formation of NETs nor did it alter formation of NETs caused by any of the conditions tested. Otherwise, it is a major distraction from the primary findings.

Reply: *We modified accordingly and moved all data related to NETs in the supplementary material (supplemental Figure S9 and S10).*

2. The flow cytometry data showing neutrophil SSC and FSC are non-specific and should not be attributed to a specific process in the absence of testing for that process directly (e.g., degranulation, line 214). I recommend moving all of these data to supplementary material and removing any mention of these morphological changes from the main text. They are not useful.

Reply: We removed all in vivo data related to cellular size and granularity from the manuscript, while the in vitro data clearly show an effect of sUA on cellular size and granularity in human neutrophils (Figure 5E-5G, and 7F-7G).

3. Figures 1-3: please define “Before” and “After” in the legends. I assume this means before and after the indicated treatment? This is somewhat confusing on cursory read, because the “before” condition never receives/or underwent the indicated treatment.

Reply: We modified the graphs in Figures 1-3 for better understanding accordingly.

4. Figure 4G: please clarify that the indicated cell surface molecules are measured on cells in blood and not simply in blood. Also, were these cells gated on Ly6G as in Figure 1E (for neutrophils as stated in the Figure 4G legend)? Please clarify.

Reply: Neutrophils were identified in the blood as CD45+CD11b+Ly6G+Ly6C- and from this neutrophil population, we determined the surface expression of CXCR4 and CD62L and CXCR2 (Figure 2, 3 and 4). Similar gating strategies were applied in Figure 1 as well as all other mouse experiments.

5. A reduction in the gene encoding Cybb should impact the ability of mature neutrophils to produce superoxide only after extended treatments of RNA synthesis inhibitors (> 1h). But not prior to that time point b/c the cells have sufficient premade stores of the protein. This has been previously reported and tested extensively.

Reply: We agree that reduced Cybb mRNA levels in sUA-treated human neutrophils (Figure 6E) are unlikely to immediately affect superoxide production, as neutrophils retain preformed NOX2 protein stores. The Cybb mRNA downregulation observed in our study due to sUA likely reflects an early transcriptional response rather than an acute loss of NOX2 activity. However, blocking NOX with diphenyleneiodonium (DPI) did not reduce superoxide production in human neutrophils cultured with E.coli compared to medium or E.coli alone (Figure 6I) but significantly reduced ROS production (Figure 6D and 6J). This would suggest that preformed NOX2 protein stores may not affect superoxide production at least in our hands, although Cybb mRNA expression is reduced.

Minor comments/edits

Line 51. “Depleting UA partially restored neutrophil dysfunction.” Should function not dysfunction.

Reply: We modified accordingly (Line 51).

Line 77. “In high-risk patients where neutrophil immunity to host defense is impaired.” Syntax issue. Please revise to “In high-risk patients where neutrophil function in host defense is impaired, ...”

Reply: We modified accordingly (Line 77).

Line 91. “...whether the sUA-mediated immune dysfunction in neutrophils counteracts the host defense...” Replace counteracts with “impairs” or “alters”, or something similar.

Reply: We modified accordingly (Line 91).

Line 222. Extend should be extent

Reply: We modified the text on the RNA-sequencing data (Line 257-282).

Line 234. Remove beads from “Dextran beads”. This is simply 10k MW dextran.

Reply: *We modified accordingly (Line 288, 290, 292).*

Line 933. “...or CKD serum eliminated from uric acid with...”. Syntax issue. Please revise to “or CKD serum treated with rasburicase to remove uric acid...”

Reply: *We modified accordingly (line 1172).*

Line 926. All surface receptor expression data should refer to “on” neutrophils rather than “in” neutrophils.

Reply: *We modified accordingly (line 1164).*

Line 934. LPS of fMLP should be LPS or fMLP

Reply: *We modified accordingly (Line 1172).*

Reviewer #2 (Remarks to the Author):

Patients with chronic kidney disease (CKD) often develop hyperuricemia, which may impair immune function. This study shows that elevated soluble uric acid (sUA) suppresses neutrophil phagocytosis, ROS, and bacterial killing. Using mouse models of endotoxemia and CLP-induced sepsis, they demonstrate that hyperuricemia, both with and without CKD, worsens infection outcomes by increasing systemic inflammation, peritoneal bacterial load, and kidney injury. CKD-associated hyperuricemia further elevated neutrophil counts, size, degranulation, cytokine levels, and renal immunothrombosis. Treatment with the uric acid-lowering drug febuxostat reduced sUA levels, kidney dysfunction, peritoneal bacterial burden, and inflammatory markers across multiple compartments. In human studies, neutrophils from CKD and end-stage kidney disease patients showed reduced phagocytic activity and ROS. Exposing healthy neutrophils to serum from CKD patients reproduced this dysfunction, which was partially reversed by rasburicase, an enzymatic uricase. These findings suggest that sUA contributes to neutrophil dysfunction and impaired host defense in CKD and that urate-lowering therapy may help restore immune competence.

Comments:

- Although the models are well described in the methods, a concise explanation in the results (i.e., which diets induce HU vs CKD+HU, with or without kidney injury) would improve readability. Clarify what “before” and “after” mean in this context.

Reply: *For better clarification, we modified the text for Figure 1 and methods (Line 541-553) accordingly.*

- While it's appreciated that the authors distinguish bacterial sepsis from endotoxemia, note that CLP is variable and lacks reproducibility. Consider briefly mentioning alternative sepsis models (e.g., cecal slurry, humanized bacterial infection models). Also, the absence of antibiotics in the CLP model reduces clinical relevance and should be acknowledged.

Reply: *We have now included a comment in the discussion section (Line 508-517). While the suggestion to include antibiotics is interesting, we chose not to incorporate this because unlike in the*

LPS- and PEP-induced inflammation models, the use of antibiotics in the CLP-induced sepsis model would significantly alter host-pathogen dynamics, would have eradicated all gut bacteria necessary to induce bacterial sepsis, and ultimately masked the immunomodulatory effects of hyperuricemia that were the primary focus of our investigation. Moreover, our patients did not receive antibiotics.

- The term "endotoxemia" typically refers to LPS-induced inflammation, not peptidoglycan. Please clarify whether PEP-induced inflammation is better described as a sterile immune challenge or inflammatory mimic of gram-positive sepsis, rather than "endotoxemia."

Reply: *We agree with the reviewer and have modified the text accordingly in referring to LPS-induced endotoxemia and PEP-induced gram-positive sepsis.*

- Inclusion of survival data after febuxostat treatment would strengthen the translational impact.

Reply: *Thank you for this valuable comment. Febuxostat is an approved drug widely used in clinical practice to reduce serum UA levels in patients with gouty arthritis. It has also been evaluated in animal models and several large multicenter RCTs involving patients with chronic kidney disease, without reports of major adverse events or toxicity. In our study, febuxostat was well tolerated in mice, and we did not observe any treatment-related adverse events and mortality. These findings, consistent with the established clinical safety profile of febuxostat, support its safe use in both experimental and clinical settings.*

We agree that survival after sepsis would represent a valuable translational endpoint. However, our local ethical authorities did not approve lethal sepsis models in mice with concomitant chronic kidney disease due to the burden where survival is the primary endpoint. For this reason we employed non-lethal sepsis models for 24 hours.

- Using side scatter (SSC) to assess degranulation is not optimal. More specific markers (e.g., CD63, CD66b) or functional granule assays would provide better mechanistic insight.

Reply: *As suggested by Reviewer 1, we have removed the FSC and SSC data from the mouse experiments. However, we retained these measures in the human neutrophils, where they provide mechanistic information (Figure 5 and 7).*

- The RNA-seq section lacks functional interpretation. Rather than stating that sUA alters gene expression and cell size, specify which pathways are affected (e.g., downregulation of migration, calcium signaling, cytoskeletal remodeling, bactericidal responses). Distinguish between phenotypic and mechanistic observations.

Reply: *We repeated the RNA-seq analysis with new samples to increase the number of samples per group and provide more detailed gene expression profiles in human neutrophils (supplemental Figure S6 and S7, and Line 254-282).*

- Overall, the study presents strong phenotypic data (impaired ROS and phagocytosis) but lacks deeper mechanistic exploration, this should be acknowledged or clarified.

Reply: *We agree with the reviewer. Teasing apart one particular pathway of action is difficult if soluble uric acid rather affects multiple neutrophil effector functions including bacterial killing, phagocytosis and ROS production, which do not share a common pathway. Our in vitro data show that intracellular soluble UA (Figure 5M) does not affect superoxide production but significantly reduces ROS production due to downregulation of NOX2 (Figure 6E, 6F) and dedranulation (Figure 5F and 5G) independent of intracellular MPO expression (Figure 6G), reduces bacterial killing (Figure 5K), and impairs phagocytosis (Figure 5H-5J) by downregulating TLR4 expression (Figure 6E) in human*

neutrophils. Together, our data suggest that soluble UA disrupts the phagocytosis-coupled NOX2–degranulation axis, where impaired TLR4 signaling reduces phagocytic activation, leading to reduced NOX2 expression and degranulation, resulting in insufficient ROS production and compromised bacterial killing (Line 52-55, and 519-524).

- The UA levels in patient samples are reported in Table 1 but not clearly stated in the results. Please summarize patient UA levels in the results and state how much blood was collected for these assays.

Reply: We modified Table 1 (Line 913) accordingly.

- The rationale for using febuxostat in mice and rasburicase in human neutrophil assays should be explained (pharmacologic differences, in vitro applicability).

Reply: We used febuxostat (a xanthine oxidase inhibitor) in mice because it is approved and effective for long-term urate-lowering therapy, as demonstrated in large multicenter RCTs. In contrast, rasburicase (a UA depleting agent) is approved only for short-term use in patients with tumor lysis syndrome and is not validated in patients with chronic kidney disease, making it unsuitable for in vivo application in our in vivo model. Our in vitro experiments with rasburicase were designed as proof-of-concept studies because only rasburicase can effectively deplete UA in the medium and patient serum, allowing us to directly assess the role of sUA. We have now included in vitro data using febuxostat (Figure 5L-5R, 6H) and as expected, no effects on neutrophil functions were observed compared with sUA alone because febuxostat inhibits UA production but does not deplete existing UA like rasburicase as mimicked in vitro.

- Are circulating neutrophil numbers different between hyperuricemic patients and healthy individuals? If not assessed, this limitation should be acknowledged.

Reply: Circulating neutrophil numbers did not differ between the groups. We included the neutrophil counts now in Table 1 (Line 913).

- The manuscript states reduced CXCR2 and increased aged neutrophils in CKD/ESKD patients, but the data show no statistical difference. These claims should be corrected for accuracy.

Reply: We modified the text to Figure 7 (line 413-422) accordingly.

- The killing assay methods are insufficiently detailed. Clarify:
 - o Were you measuring extracellular bacteria only, or total (extra + intracellular)?
 - o How was phagocytosis distinguished from killing?
 - o How was bacterial uptake confirmed prior to measuring killing?
 - o Why were neutrophils stimulated with LPS before E. coli exposure?
 - o Could the neutrophil response to LPS vs E. coli differ by disease state?

Reply: In the bacterial killing assay, which is widely used, we quantified extracellular CFU in the supernatants after neutrophils were removed by centrifugation to assess bacterial survival after 24 hours. Higher CFU values indicate greater E.coli survival, and conversely, reduced neutrophil-mediated bacterial killing (Figure 5K).

Phagocytosis was assessed separately from killing using Dextran, phrodo-labeled E.coli bioparticles or BFP-labeled live E.coli by flow cytometry. It is well established that during sepsis, neutrophils primarily kill bacteria after phagocytosis, when engulfed microbes are exposed to ROS, proteases and antimicrobial paptides within the phagolysosome. Non-phagocytic killing mechanisms exist but are typically relevant for larger pathogens are unlikely to play a major role with E.coli.

LPS stimulation was used to mimic the primed inflammatory state of neutrophils during sepsis and to enhance TLR4-mediated responses before bacterial challenge, consistent with established protocols. Our data suggest that in CKD-related hyperuricemia, neutrophils display reduced responsiveness to both stimuli, consistent with an overall hyporesponsive or “paralyzed” phenotype rather than a selective defect in either LPS or bacterial recognition.

- In Fig 6A, the CFU values ($\sim 10^8/\mu\text{L}$) appear extremely high given the reported MOI of 0.05 per 10^6 neutrophils. This discrepancy should be rechecked for accuracy or clarified in the methods.

Reply: *The high CFU values in Figure 5K refer to the E.coli without neutrophils (E.coli only), which was included as a control to show that E.coli concentration stays consistent over time without the presence of neutrophils. In contrast, CFU values after stimulation with neutrophils were substantially lower, ranging from approx. 2×10^3 to 3×10^4 , consistent with effective bacterial killing.*

- In Figure 1B, it is unclear what “before” and “after” refer to—presumably before and after the acidogenic diet. This should be clarified both in the figure legend and results text.

Reply: *We modified the Figures 1, 2, and 3, as well as the figure legends accordingly. “Before” refers to Day 23 prior to sepsis induction and “After” refers to Day 24 (24 hours) after sepsis induction.*

- In Figure 1C, 1K–L, serum uric acid is elevated in both “before” and “after” LPS groups, but only the “after” group develops kidney dysfunction. This discrepancy needs to be explained—possibly related to cumulative exposure or duration?

Reply: *We thank the reviewer for this important observation. Yes, this is correct because serum uric acid levels only increased in mice fed a chow diet with inosine to induce hyperuricemia (HU) compared to healthy mice before PBS and LPS injection or CLP surgery. Injecting mice with LPS or performing CLP did not alter serum uric acid levels, as shown in Figure 1B and 1K. Please also see the data in supplemental Figure S1, where we show baseline characteristics of the model of hyperuricemia with and without chronic kidney disease prior to sepsis/inflammation induction. The observed effects in kidney function, where creatinine rises only after LPS or CLP exposure, is consistent with clinical observations. Sepsis is a well-known trigger of kidney injury. In our model, hyperuricemic mice exhibit greater susceptibility to LPS- and CLP-induced kidney dysfunction, reflected in elevated plasma creatinine levels post sepsis induction (Figure 1C and 1K). We agree with the reviewer that this likely reflects the cumulative impact of hyperuricemia combined with a second hit (sepsis), rather than hyperuricemia alone.*

- Figure 1S kidney section results are labeled incorrectly in the results text; the figure reference should be corrected.

Reply: *We thank the reviewer for alerting us and modified the text accordingly (Line 100-115).*

- Figure 2E description of neutrophil markers is inaccurate. Results show no change in CD62L, reduced CXCR4, and increased CXCR2 in CKD mice during endotoxemia, not as stated in the text. Please revise for accuracy and interpret significance appropriately.

Reply: *We modified accordingly (Line 162-164).*

- In Figure 1M, bacterial clearance is shown to be impaired in CKD mice, but CFUs should be quantified to substantiate the claim.

Reply: We now included data on the colony area of bacteria growth in the CLP mouse model (Figure 1N).

- Supplemental Figure 1, mouse models are described clearly here, but a brief summary should be included in the main results to aid clarity.

Reply: We modified accordingly (Line 100-115).

- Evidence of kidney injury in HU mice (e.g., creatinine, fibrosis) is currently only in supplemental figures, this should be moved to the main figures given its importance.

Reply: We chose not to include these model characterization data in the main figures in order to maintain the focus of the story. We have extensively characterized and validated our mouse model of hyperuricemia with and without kidney disease (PMID 32938648, 32561569, 35303071, 35203277, 37001603). Our previous work, as well as multiple large multicenter RCTs on ULT, has consistently shown that hyperuricemia alone does not cause kidney injury and does not contribute to CKD progression. Upon request, we are happy to relocate these data in the revised manuscript.

- Figure 7/Table 1, CKD and ESKD patients in Table 1 have serum UA levels below the defined hyperuricemia range (9–14 mg/dL). Consider addressing this inconsistency and clarifying whether any of these patients had infections, to support the clinical relevance of findings.

Reply: Hyperuricemia is clinically defined by serum UA levels above 6.5 mg/dL in men and 6.0 mg/dL in women, and aligns with the levels observed in our patient cohort (Table 1). The patients enrolled in this study did not have infections.

This study addresses an important and understudied aspect of immune dysfunction in CKD-related hyperuricemia, offering compelling evidence that soluble uric acid impairs neutrophil-mediated host defense during sepsis. The data are promising and the translational relevance is high. However, the manuscript is significantly hindered by unclear figure legends, inconsistent and sometimes inaccurate interpretation of results, incomplete methodological descriptions, weak mechanistic insight, and underdeveloped or clinically limited models (e.g., lack of antibiotics, questionable hyperuricemia definitions in patient cohorts). These issues collectively detract from the clarity, rigor, and overall impact of the work. Addressing these concerns will be essential to strengthen the conclusions and improve the manuscript's contribution to the field.

Reply: We thank the reviewer for the positive evaluation and have revised the manuscript accordingly to improve clarity and consistency throughout the text. Regarding the animal models, we disagree with the characterization of them as “underdeveloped or clinically limited”. Our animal model is the only animal model of hyperuricemia with and without chronic kidney disease currently available that closely mimics the human pathophysiology (PMID 32938648). Importantly, our model achieves serum UA levels above 6.5 mg/dL, which is clinically defined as hyperuricemia and aligns with the levels observed in our patient cohort. We therefore believe that this model offers robust physiological and translational relevance.

While the suggestion to include antibiotics is interesting, we chose not to incorporate this because unlike in the LPS- and PEP-induced inflammation models, the use of antibiotics in the CLP-induced sepsis model would significantly alter host-pathogen dynamics, would have eradicated all gut bacteria necessary to induce bacterial sepsis, and ultimately masked the immunomodulatory effects of hyperuricemia that were the primary focus of our investigation. Moreover, our patients did not receive antibiotics.

Reviewer #3 (Remarks to the Author):

The authors present evidence that hyperuricemia may exacerbate inflammation due to infection or endotoxemia by altering the ability of neutrophils to kill bacteria. Specifically, they use an interesting model of hyperuricemia due to liver-specific knockout of Glut9 in which hyperuricemia develops in response to inosine but CKD only develops if the diet is acidified (thought to be related to crystalluria or intrarenal crystal). They first show that either LPS or cecal puncture/sepsis worsens AKI and bacterial cultures in the peritoneum in hyperuricemic mice compared to healthy controls. They also show that their hyperuricemia CKD model is also exacerbated by LPS or cecal puncture, and in these models the administration of febuxostat is associated with less sepsis and inflammation, and better creatinine. They then show that soluble uric acid affects neutrophil phagocytosis and oxidant release and impairs bacterial killing, but not NET formation. Finally, they take human neutrophils from healthy subjects or with CKD and show that CKD is associated with higher uric acid levels and worse neutrophil function, and that reducing uric acid with rasburicase can improve neutrophil function.

Overall this is very nice work, and the authors are to be congratulated. However, there are some very important experiments missing.

Reply: *We thank the reviewer for his/her positive evaluation of our manuscript.*

1. The most important study is to determine if hyperuricemia worsens sterile inflammation. Indeed, the authors seem to imply uric acid might be protective here, but there is no convincing evidence presented. I suggest the authors use a model such as carageenan induced peritonitis. I believe hyperuricemia will make kidney disease worse, as determined with and without febuxostat.

Reply: *We have previously demonstrated that hyperuricemia suppresses sterile inflammation, e.g. in gouty arthritis, by impairing neutrophil migration (PMID 35303071) and altering monocyte function (PMID 32561569), suggesting an immunoregulatory (suppressive) role of sUA.*

In the current study, we aimed to address the question: whether due to this suppressed immune system, hyperuricemia contributes to the increased risk of infection in CKD patients, a phenomenon referred to as secondary immunodeficiency related to kidney disease. Our findings support this hypothesis. Specifically, while hyperuricemia alone does not cause kidney dysfunction, it makes mice more susceptible to LPS, PEP or CLP exposure, despite comparable UA levels before and after intervention (Figure 1B and 1L). However, the observed effects in kidney function, where creatinine rises only after LPS or CLP exposure, is consistent with clinical observations. Sepsis is a well-known trigger of acute kidney injury. In our model, hyperuricemic mice exhibit greater susceptibility to LPS- and CLP-induced kidney dysfunction, reflected in elevated plasma creatinine levels post-intervention (Figure 1C and 1K), which was even more pronounced in mice with chronic kidney disease (Figure 2, 3, Supplementary Figure S4).

Moreover, previous experimental data from our group have shown that hyperuricemia alone does not cause kidney injury nor contribute to CKD progression (PMID 32938648), which has been confirmed by large multicenter RCT on urate-lowering therapy (PMID 32313212). The human findings align with our current data, where kidney injury was only observed following LPS or CLP exposure (Figure 1, Supplementary Figure S1), similar to human experiencing sepsis-induced acute kidney injury. We feel that using an additional model of carageenan-induced peritonitis would not provide additional insights beyond what we have already addressed in the LPS, PEP and CLP models.

2. The authors also show that asymptomatic hyperuricemia is not so asymptomatic if endotoxin or sepsis occurs, and this is associated with worsening creatinine (AKI) and neutrophilia, etc. Again, this should be shown to be due to the uric acid by doing studies with febuxostat.

Reply: We have clarified the confusion about asymptomatic and symptomatic hyperuricemia in a recent review article (PMID 37261000). Asymptomatic hyperuricemia means “soluble UA” and mainly occurs in the context of CKD due to an impaired urinary clearance. Only when UA crystallizes, hyperuricemia becomes symptomatic as in gouty arthritis, kidney stones, urolithiasis, acute and chronic UA nephropathy. Therefore, using the term “asymptomatic HU” is correct. Please also see our comment above. Asymptomatic hyperuricemia by itself does not cause kidney injury (Figure 1). Rather it is only upon exposure to LPS, PEP and CLP that lead to an increase in creatinine levels and heightened systemic inflammation. To address whether these detrimental effects are mediated by hyperuricemia, we used febuxostat in hyperuricemic mice with and without chronic kidney disease. Febuxostat effectively lowered serum UA levels and improved host defense during LPS, PEP and CLP exposure in hyperuricemic mice with and without chronic kidney disease. This was associated with reduced neutrophilia, improved bacterial killing and preserved kidney function (Figure 2, Figure 4, supplementary Figure S4). To address the reviewer’s concern, we now included data also of febuxostat treatment in mice with hyperuricemia alone (supplementary Figure S3) and observed similar effects.

3. There is some data showing histology that hyperuricemia is affecting SMA or fibrin deposition. Can the authors do specific staining of the kidneys for inflammation, fibrosis, SMA and fibrin? We need to know how the kidneys are being affected.

Reply: In line with our previous findings (PMID 32938648, 32561569, 35303071) and clinical data, asymptomatic hyperuricemia alone does not cause kidney injury nor contribute to the progression of chronic kidney disease. Such data are already included in supplemental Figure S1 demonstrating that hyperuricemia alone (HU) does not impair kidney function, as indicated by a similar GFR, creatinine levels, no kidney injury (PAS stain, mRNA levels of KIM-1) and fibrosis (Pirco sirius Red stain, mRNA levels of Fibronectin1) compared to healthy mice. Only mRNA levels of TNF α and IL-6 are increased. However, placing mice on an acidogenic diet with inosine causes chronic kidney disease (HU+CKD) as indicated by impaired kidney function due to injury, inflammation and fibrosis.

4. The model of hyperuricemia is quite unique and many investigators do not believe this is a good model as it affects liver processing of uric acid. It might be ideal to show at least in one other model of hyperuricemia that these findings hold.

Reply: We appreciate the reviewer’s comment. In humans, UA is primarily synthesized in the liver, intestine and vascular endothelium as the final product of the purine metabolism. Hyperuricemia can arise from multiple factors including chronic kidney disease (primary cause, 80-90% of cases), purine- and fructose-rich diets, alcohol consumption, metabolic syndrome, diabetes, and genetic mutations. Our transgenic mouse model is based on the urate transporter Glut9 and an purine-rich diet with inosine, which leads to elevated serum UA levels (above 6.5 mg/dl), meeting the clinical definition of hyperuricemia. In addition, by placing mice on an acidogenic diet in combination with inosine, we can induce hyperuricemia and UA crystal-induced CKD in a manner that closely reflects human disease. This model has been extensively validated and characterized in our previous publications (PMID 32938648, 32561569, 35303071, 35203277, 37001603), demonstrating pathological features relevant to human disease.

Importantly, hepatic UA production is a physiological and conserved process, not an artifact of our model. The concern about “liver processing of UA” does not compromise the translational relevance of our findings; rather, it reflects the natural biology of urate synthesis shared between mice and humans.

While we understand the reviewer’s interest in additional validation, we would like to point out that other widely used animal models, including potassium oxonate administration, unilateral ureteral obstruction, or diabetic nephropathy, do not achieve clinically relevant UA levels and do not reflect the systemic, multifactorial nature of hyperuricemia with kidney disease observed in human patients. Additionally, some of these models involve acute or artificial interventions (e.g., i.p. uric acid

injections) that lack pathophysiological relevance. Given that our HU+CKD model achieves robust, sustained hyperuricemia and mimics the metabolic and kidney alterations seen in patients, we believe it currently offers the most clinically relevant model for studying UA-associated immune dysfunction in kidney injury. We respectfully suggest that the introduction of a second, less representative model would not enhance the translational validity of our conclusions.

5. Finally, incubation studies with rasburicase are quite problematic as rasburicase releases oxidants on reducing urate. These studies should be deleted. The human studies would have been much stronger by altering uric acid with XO inhibitors and documenting reduced XO activity and intracellular urate levels.

Reply: *This is an important point to clarify. Rasburicase is an effective drug to deplete UA and to our knowledge does not directly induce the release of oxidants. In contrast, the xanthine oxidase itself does generate the release of reactive oxygen species for example in macrophages, thereby contributing to the pro-inflammatory effects independent of UA (PMID 25800347). To address the reviewer's concern, we performed additional in vitro experiments also with febuxostat. As expected, the xanthine oxidase inhibitor febuxostat did not deplete UA in the medium of sUA-treated neutrophils, and therefore, neutrophil functions were still impaired similar to sUA alone. Rasburicase was simply used as proof-of-concept to directly test the impact of UA on neutrophil functions. See Figure 5L-5R, as well as Figure 6F and 6H).*

6. Uric acid is also known to form MPO- related adducts and to block peroxynitrite, and how this plays into the results shown is not so clear.

Reply: *We are with the reviewer that UA can react with MPO-derived oxidants (especially hypochlorous acid, HOCl), forming urate oxidation products and urate-MPO adducts in neutrophils. UA can also scavenges peroxynitrite and other reactive species, reducing oxidative burst but potentially limiting microbicidal ROS activity, which is relevant to our findings. We have now included a comment in the discussion (Line 474-486).*

Reviewer Comments:

> Reviewer #2 (Remarks to the Author):

The authors have substantially revised and improved the manuscript according to reviewer comments.

Reply: *We thank the reviewer for his/her positive evaluation of our manuscript.*

> **Reviewer #3 (Remarks to the Author):**

The main finding of the paper is that infection or LPS can induce a more prominent inflammatory response in hyperuricemic mice associated with renal dysfunction and that this can be improved by lowering the serum uric acid with febuxostat. While superficially the enhanced inflammatory response associated with hyperuricemia would suggest uric acid is promoting inflammation, the authors make the case that this is the reverse, based on in vitro studies that uric acid is impairing the killing of the bacteria due to effects on phagocytosis and oxidative burst. There remain several key issues that I feel should be addressed.

1. The authors actually have mixed data with uric acid showing some proinflammatory and anti-inflammatory effects, so perhaps dysfunctional is a better term.

Reply: *We believe this point requires clarification. We did not present mixed data suggesting that uric acid has pro- and anti-inflammatory effects during endotoxemia or sepsis. Rather, our data demonstrate that uric acid induces a hyperinflammatory response, characterized by increased IL-6 and IL-1beta levels, elevated neutrophil counts, and enhanced cellular activation (degranulation and reduced CD62L expression), indicating a pro-inflammatory effect in this context. At the same time, uric acid impairs effective host defense by inhibiting phagocytosis, ROS production, and bacterial killing, which is detrimental during infection. Thus, uric acid simultaneously promotes hyperinflammation while compromising antimicrobial function. Our findings are consistent with the clinical phenotype observed in patients with CKD and provide a mechanistic explanation for the increased risk of infection in these patients. We agree that the term “dysfunctional” or „immunoregulatory“ is better.*

For example, in both in vitro and in vivo studies, uric acid is associated with enhanced degranulation, which is a principle sign of neutrophil activation. They actually show data that extracellular MPO levels are high. We need to know more about this- are proteases and defensins being released in addition to MPO? This is a classic proinflammatory effect. Chemotaxis was also not measured. This should also be done.

Reply: *We agree with the reviewer that degranulation and lower CD62L expression are signs of neutrophil activation, which also accounts for the observed hyperinflammatory response, reflected by increased IL-6, IL-1beta and MPO levels and neutrophil counts. In addition to increased MPO release as sign of NET release in vivo, our in vitro data on NET formation indicate that neutrophils also release cit. histone H3 (Supplemental Figure S9, S10). We believe that additional measurements of proteases or defensins would not provide substantial further information.*

In this study, we focused on key effector functions relevant to host defense, including cell activation, phagocytosis, ROS production, bacterial killing and NET formation. We noticed increased neutrophil numbers in blood and peritoneum of hyperuricemic CKD mice compared to healthy mice after LPS, PEP and CLP (Figure 2D, 2H, 3D, 3J, Supplemental Figure S4E, S4G). This was associated with increased expression of CXCR2 (Figure 2F and 3F) and of the beta2 integrin MAC-1 in blood (see graphs below), peritoneum and spleen (data not shown here), consistent with increased neutrophil chemotaxis and recruitment. While formal chemotaxis in vitro assays were not performed, these

findings support augmented neutrophil recruitment. We did not include the MAC-1 expression data in the initial submission but would do so upon request.

In contrast, the reduction in oxidants presented is most likely from quenching of peroxynitrite with uric acid. This would impair killing.

Reply: We agree with the reviewer.

Despite hyperuricemic PMNs showing impaired killing, they may also be proinflammatory. One interesting finding by the authors is that the neutrophils are aged. One explanation could be that hyperuricemia may act as a DAMP to stimulate low grade activation of PMNs, which are relatively spent and exhausted and may not mount effective killing but are contributing to systemic inflammation. It might be interesting to do a time response to see how long it takes for a healthy PMN to change its phenotype to an aged neutrophil (CXCR4) in the presence or absence of uric acid. Then a comparison of neutrophil inflammatory responses (chemotaxis, inflammatory cytokine release) in the two states could be compared.

Reply: We agree with the reviewer that uric acid exerts pro-inflammatory effects by inducing cellular activation, cytokine release, and neutrophil recruitment, while simultaneously impairing host defense during infection (see comment above). However, we are careful with referring to hyperuricemia or soluble uric acid as DAMP because this only applies to cell activation, but not to antimicrobial functions during infection. In contrast, hyperuricemia suppresses sterile inflammation as in gouty arthritis (Blood, PMID 35303071), and does not induce kidney injury nor contribute to CKD progression (JASN, PMID 32938648). Only when soluble uric acid crystallizes, such crystals act as DAMPs causing gouty arthritis, acute and chronic UA nephropathy, urolithiasis, and kidney stone disease, as shown by us and others.

In our study, we observed fewer aged neutrophils, defined by lower CXCR4 expression, in hyperuricemic conditions, suggesting a neutrophil phenotype of enhanced mobilization rather than senescence. We propose sustained exposure to soluble uric acid promotes neutrophil activation and mobilization, ultimately leading to functional exhaustion as evident by persistent inflammation and immune paralysis in patients with kidney disease (JASN, PMID 34907031).

2. The authors hypothesize that the worse inflammation with hyperuricemia is because the uric acid is impairing bacterial clearance, so the inflammation mounts. In this scenario, the animals should die sooner from severe infection. However, the authors state that survival studies cannot be done so this important study cannot be done. However, If the authors are correct, and the proinflammatory response is from a higher bacterial load, then treating animals with antibiotics would be expected to fully correct the increased inflammatory response in hyperuricemic animals. However, if hyperuricemia is causative for a greater inflammatory response, then we should see the opposite. One way to do this is to initiate antibiotics at the time the cecal ligation model is induced ,and to compare results.

Reply: Yes, we agree with the reviewer that uric acid exerts pro-inflammatory effects, while simultaneously impairs host defense during infection (see comment above), as indicated by a higher bacterial load in Figure 1M, 1N, 3G, 3H.

While the suggestion to include antibiotics is interesting, we chose not to incorporate this because unlike in the LPS and PEP model, the use of antibiotics in the CLP-induced sepsis model would significantly reduce bacterial load due to eradicated bacteria necessary to induce host-pathogen responses and hyperinflammation, and ultimately masked the immunomodulatory effects of hyperuricemia that were the primary focus of our investigation. Moreover, our patients did not receive antibiotics. We had included a comment in the discussion section (Line 496-498). We have proven causality by treating mice with febuxostat (Figure 2, 4, Supplemental Figure S2, S4).

3. The other way to test this is with a true sterile inflammation model. I do not think the prior studies address this. Carageenan induced peritonitis only takes a few hours to induce and would be perfect (see PMID: 26786873). I would recommend doing this two ways.

One way is to create the hyperuricemic model with inosine for 23 days and then do an acute carrageenan peritonitis. The other would be to give the inosine one day before the carrageenan. This way one can get an idea for a chronic hyperuricemic state (such as in the current models presented) and a more acute model.

Reply: We thank the reviewer for this thoughtful suggestion. We would like to note, however, that we have previously demonstrated that hyperuricemia/soluble uric acid suppress „true“ sterile inflammation in gouty arthritis (Blood, PMID 35303071). Therefore, applying an additional sterile inflammation animal model such as carageenan-induced peritonitis would not provide new insights beyond what is already shown in this study (LPS, PEP and CLP) and our previous study.

Of note, feeding mice with inosine for one day does not result in increased serum uric acid levels and therefore would not accurately reflect either experimental hyperuricemia nor the clinical condition of persistent hyperuricemia in patients with CKD.

4. The human PMN studies are problematic since kidney dysfunction and diabetes are known to induce PMN dysfunction, and so evidence that this is due to hyperuricemia vs other mechanisms affecting neutrophils. There are CKD patients with normal uric acid levels (only 50 percent are hyperuricemic at time of initiation of dialysis) so the right control would be nondiabetic CKD with similar GFR but different uric acid levels. The authors also do not understand how rasburicase works. When it degrades uric acid, it produces oxidants (primarily hydrogen peroxide) so it is not a good way for testing uric acid in an in vitro system. I would probably delete this whole section unless it is done with the correct controls.

Reply: We appreciate the reviewer's concerns and would like to clarify several points. Large studies, including NHANES (2007-2010) in the US and the German CKD cohort (2010-2012), showed that nearly 80% of patients with advanced CKD are hyperuricemic at the time of initiation of dialysis, rather than 50%. These data support the notion that kidney disease is the primary cause of hyperuricemia. While CKD patients with normal uric acid levels do exist, they present a minority. Importantly, all patients included in our study had both hyperuricemia and CKD, ensuring internal consistency of the cohort despite diverse underlying kidney disease of the patients (Table 2), ranging from genetic, metabolic and autoimmune to drug-induced.

We do not claim that uric acid is the sole contributor to neutrophil dysfunction in CKD; rather, it is one of several uremia-related factors (see introduction, Line 78-86) contributing to the secondary immunodeficiency. Importantly, hyperuricemia is clinically treatable with agents like febuxostat (FDA approved), whereas other uremic proteins (e.g. FGF23, resistin, leptin) currently lack targeted therapies and would require treatment of CKD itself with e.g. SGLT2, MRAs, or RAS inhibitors.

We note that hyperuricemia is also associated with diabetes and metabolic syndrome, and that hyperuricemia, diabetes, obesity, hypertension, autoimmune diseases, cancer and CKD can influence neutrophil functions (see introduction Line 78-86). The mechanisms linking uric acid to diabetes are complex and multifactorial, involving insulin resistance, purine intake, uric acid handling in kidney and gut, drugs, and oxidative stress. Investigating these interactions in depth would require separate, dedicated studies in mice and patients (groups: diabetic and non-diabetic, with and without hyperuricemia, with and without CKD, and with and without infection), as diabetes occurs mainly in patients without CKD and hyperuricemia. Importantly, all patients in our current study were selected based on the presence of both CKD and hyperuricemia (Table 1 and 2), independent of diabetes and other comorbidities, to specifically interrogate the contribution of elevated uric acid to neutrophil dysfunction in CKD during infection. While diabetes may influence neutrophil function via overlapping but also distinct pathways, disentangling these effects is beyond the scope of this study and would require larger, mechanistically distinct cohorts, in vivo and in vitro studies. Our focus remains on establishing proof-of-concept that hyperuricemia contributes to neutrophil dysfunction in CKD during sepsis, providing a clinically treatable target. We added a comment in the discussion Line 426-430.

Regarding the use of rasburicase, we are fully aware that uric acid degradation can generate hydrogen peroxide. However, rasburicase was used as a mechanistic tool to selectively deplete soluble uric acid under controlled conditions in vitro. We did not observe increased oxidative stress in rasburicase-treated neutrophils compared to untreated neutrophils (Figure 6H). Rasburicase is approved only for short-term use in patients with tumor lysis syndrome and gouty arthritis (please see our paper in Blood, PMID 35303071, where we used rasburicase), and is not validated in patients with CKD, making it unsuitable for in vivo application in our in vivo sepsis model. Currently, no effective uric acid depleting agents other than rasburicase exist. Pegloticase is not available in the EU. The observed in vitro effects were consistent with those obtained using alternative approaches, including the XO1 febuxostat in vivo (Figure 2I-2S, 4, supplemental Figure S2), which however does not deplete soluble uric acid in vitro (Figure 5L-5R), supporting the conclusion that soluble uric acid itself modulates neutrophil functions. Thus, the rasburicase experiments do not confound but rather strengthen the mechanistic interpretation (proof-of-concept) when viewed in the context of the entire dataset. We added a comment in the discussion Line 490-494.

> Additional reviewer advice on Reviewer #3 comments

1. Regarding the request for an added sterile inflammation model (carrageenan peritonitis):

The suggestion is scientifically reasonable in principle: a sterile peritonitis model can help distinguish between inflammation driven by pathogen-associated stimuli versus endogenous danger signals such as uric acid.

However, in the context of this manuscript, I agree with the editorial assessment that adding a new sterile model is not essential and may not be appropriate for this revision stage.

The central question of the study is how soluble uric acid impairs neutrophil-mediated host defense during infection. The authors already use two infection-related models (LPS/PEP inflammatory challenges and CLP bacterial sepsis), and the main mechanistic conclusions relate to antimicrobial functions rather than sterile inflammation. Adding an entirely new animal model would substantially expand the scope without clearly improving the core message.

Reply: *We thank the reviewer for this comment that applying an additional animal model such as carrageenan-induced peritonitis would not provide new insights beyond what is already shown in this study (LPS, PEP and CLP) and what we have previously demonstrated during sterile inflammation in gouty arthritis (Blood, PMID 35303071).*

We also wish to clarify that we do not consider soluble uric acid as a general danger signal like it's crystalline form, as it's effects are context-dependent (see comment above).

2. Regarding the concern about rasburicase in the human neutrophil studies:

Reviewer 3 is correct that rasburicase generates hydrogen peroxide as a byproduct of uric acid degradation. Because of this, its use can introduce confounding oxidative effects in vitro. The reviewer's point is scientifically valid and worth addressing. That said, this does not necessarily mean the entire human PMN section must be deleted. Instead, the authors could: acknowledge this limitation directly, clarify why rasburicase was chosen (clinical relevance, necessity of enzymatic UA degradation instead of synthesis inhibition), provide appropriate controls or interpret the data with caution. The rasburicase concern does not invalidate the entire dataset, but it does warrant clarification and a toned-down interpretation.

Reply: *Regarding the use of rasburicase, we are fully aware that uric acid degradation with rasburicase can generate hydrogen peroxide (see comment above). However, rasburicase was used as a mechanistic tool to deplete soluble uric acid under controlled conditions in vitro. We did not observe increased oxidative stress in rasburicase-treated neutrophils compared to untreated neutrophils (Figure 6H). Rasburicase is approved only for short-term use in patients with tumor lysis syndrome and gouty arthritis, and is not validated in patients with CKD, making it unsuitable for in vivo application in our in vivo model. Currently, no effective uric acid depleting agents other than rasburicase exist. Pegloticase is not available in the EU. The observed effects were consistent with those obtained using alternative approaches, including the XO1 febuxostat in vivo (Figure 2I-2S, 4, supplemental Figure S2), which however does not deplete soluble uric acid in vitro (Figure 5L-5R), supporting the conclusion that soluble uric acid itself modulates neutrophil functions. Thus, the rasburicase in vitro experiments do not confound but rather strengthen the mechanistic interpretation (proof-of-concept) when viewed in the context of the entire dataset. We added a comment in the discussion Line 490-494.*

3. Regarding the reviewer's concerns about CKD vs hyperuricemia in human samples:

It is true that CKD and diabetes independently cause neutrophil dysfunction, and that the current patient cohort does not isolate hyperuricemia per se. This is a reasonable critique. The authors can address this by: clarifying that their human data demonstrate an association rather than causal attribution to UA alone, adding a discussion acknowledgement that matched CKD patients with differing UA levels would be the ideal comparator, but was not available. This is a limitation but does not undermine the overall value of including human data, in my opinion.

Reply: *We thank the reviewer for this comment. We agree that CKD and diabetes can independently affect neutrophil functions and that our human cohort cannot fully isolate the effects of hyperuricemia per se. Large studies, including NHANES (2007–2010) and the German CKD cohort (2010–2012), indicate that ~80% of patients with advanced CKD are hyperuricemic at dialysis initiation, supporting that CKD is the primary cause of hyperuricemia. While CKD patients with normal uric acid exist (due to genetic factors, purine & fructose diet, alcohol intake, drugs, increased cell turnover, altered uric acid handling), they represent a clear minority (5-10%). That is why all patients in our study had both CKD and hyperuricemia, ensuring internal consistency. Importantly, our in vivo data show that „isolated“ hyperuricemia alone induces hyperinflammation and impairs host defense (Figure 1), which is further aggravated in combination with CKD (Figure 2-4), providing clear evidence. We acknowledge that hyperuricemia is associated with diabetes and metabolic syndrome; however, not all patients with diabetes develop hyperuricemia. This depends on several factors such as diabetes*

type, kidney function, drugs, and other metabolic comorbidities such as obesity and hypertension. The mechanisms linking uric acid to diabetes are complex, involving insulin resistance, purine intake, uric acid handling in kidney and gut, renal uric acid clearance, and oxidative stress. Dissecting these interactions in depth would require separate, dedicated studies using animal models and patients cohorts (groups: diabetic and non-diabetic, with and without hyperuricemia, with and without CKD, with and without infection), which is beyond the scope of this study. Similarly, other comorbidities such as autoimmune disease, cancer, hypertension and obesity can also influence neutrophil functions independently or in combination with CKD and hyperuricemia, as mentioned in the introduction Line 78-86 and discussion Line 426-430. Our focus was to establish proof-of-concept that hyperuricemia contributes to neutrophil dysfunction in CKD during sepsis without dissecting each comorbidity. Importantly, our in vivo febuxostat experiments demonstrate causality (Figure 2, 4, supplemental Figure S2, S4), complementing the associative human data. Researchers and clinicians are aware that all published human data are associative unless patients with genetic-related diseases (e.g. inborn errors of immunity or metabolism) are included or a clinical trial has proven or disproven causality.